# Stochastic Variance-Reduced Newton: Accelerating Finite-Sum Minimization with Large Batches

**Michał Dereziński**                                                                    *derezin@umich.edu*
*Department of Electrical Engineering and Computer Science*
*University of Michigan*

Reviewed on OpenReview: *https://openreview.net/forum?id=dzQCRHKRdC*

## Abstract

Stochastic variance reduction has proven effective at accelerating first-order algorithms for solving convex finite-sum optimization tasks such as empirical risk minimization. Incorporating second-order information has proven helpful in further improving the performance of these first-order methods. Yet, comparatively little is known about the benefits of using variance reduction to accelerate popular stochastic second-order methods such as Subsampled Newton. To address this, we propose Stochastic Variance-Reduced Newton (SVRN), a finite-sum minimization algorithm that provably accelerates existing stochastic Newton methods from $O(\alpha \log(1/\epsilon))$ to $O\left(\frac{\log(1/\epsilon)}{\log(n)}\right)$ passes over the data, i.e., by a factor of $O(\alpha \log(n))$, where $n$ is the number of sum components and $\alpha$ is the approximation factor in the Hessian estimate. Surprisingly, this acceleration gets more significant the larger the data size $n$, which is a unique property of SVRN. Our algorithm retains the key advantages of Newton-type methods, such as easily parallelizable large-batch operations and a simple unit step size. We use SVRN to accelerate Subsampled Newton and Iterative Hessian Sketch algorithms, and show that it compares favorably to popular first-order methods with variance reduction.

## 1 Introduction

Consider a convex finite-sum minimization task:

$$\text{find} \quad \mathbf{x}^* = \operatorname*{argmin}_{\mathbf{x} \in \mathbb{R}^d} f(\mathbf{x}) \quad \text{for} \quad f(\mathbf{x}) = \frac{1}{n} \sum_{i=1}^{n} \psi_i(\mathbf{x}). \tag{1}$$

This optimization task naturally arises in machine learning through empirical risk minimization, where $\mathbf{x}$ is the model parameter vector and each function $\psi_i(\mathbf{x})$ corresponds to the loss incurred by the model on the $i$-th element in a training data set (e.g., square loss for regression, or logistic loss for classification). Many other optimization tasks, such as solving semi-definite programs and portfolio optimization, can be cast in this general form. Our goal is to find an $\epsilon$-approximate solution, i.e., $\tilde{\mathbf{x}}$ such that $f(\tilde{\mathbf{x}}) - f(\mathbf{x}^*) \leq \epsilon$.

Classical iterative optimization methods (such as gradient descent and Newton's method) use first/second-order information of function $f$ to construct a sequence $\mathbf{x}_0, \mathbf{x}_1, ...$ that converges to $\mathbf{x}^*$. However, this does not leverage the finite-sum structure of the problem. Thus, extensive literature has been dedicated to efficiently solving finite-sum minimization tasks using stochastic optimization methods, which use first/second-order information of randomly sampled component functions $\psi_i$, that can often be computed much faster than the entire function $f$. Among first-order methods, variance-reduction techniques such as SAG (Roux et al., 2012), SDCA (Shalev-Shwartz & Zhang, 2013), SVRG (Johnson & Zhang, 2013), SAGA (Defazio et al., 2014), Katyusha (Allen-Zhu, 2017) and others (Frostig et al., 2015; Konecný et al., 2015; Allen-Zhu & Yuan, 2016), have proven particularly effective. One of the most popular variants of this approach is Stochastic Variance-Reduced Gradient (SVRG), which achieves variance reduction by combining frequent stochastic gradient queries with occasional full batch gradient queries, to optimize the overall cost of finding an $\epsilon$-approximate solution, where the cost is measured by the total number of queries to the components $\nabla \psi_i(\mathbf{x})$.

Many stochastic second-order methods have also been proposed for solving finite-sum minimization, including Subsampled Newton (Erdogdu & Montanari, 2015; Roosta-Khorasani & Mahoney, 2019; Bollapragada et al., 2018; Berahas et al., 2020; Dereziński & Mahoney, 2019) Newton Sketch (Pilanci & Wainwright, 2016; 2017; Dereziński et al., 2021), and others (Kovalev et al., 2019; Moritz et al., 2016; Tripuraneni et al., 2018; Mokhtari et al., 2018; Gupta et al., 2021). These approaches are generally less sensitive to hyperparameters such as the step size, and they typically query larger random batches of component gradients/Hessians at a time, as compared to first-order methods. The larger queries make these methods less sequential, allowing for more effective vectorization and parallelization.

A number of works have explored whether second-order information can be used to accelerate stochastic variance-reduced methods, resulting in several algorithms such as Preconditioned SVRG (Gonen et al., 2016), SVRG2 (Gower et al., 2018) and others (Gower et al., 2016; Liu et al., 2019). However, these are still primarily stochastic first-order methods, highly sequential and with a problem-dependent step size. Comparatively little work has been done on using variance reduction to accelerate stochastic Newton-type methods for convex finite-sum minimization (see discussion in Section 2.3). To that end, we ask:

*Can variance reduction accelerate local convergence of Stochastic Newton*
*in convex finite-sum minimization?*

We show that the answer to this question is positive. The method that we use to demonstrate this, which we call Stochastic Variance-Reduced Newton (SVRN), retains the positive characteristics of second-order methods, including easily parallelizable large-batch gradient queries, as well as minimal hyperparameter tuning (e.g., accepting a unit step size). We prove that, when the number of components $\psi_i$ is sufficiently large, SVRN achieves a better parallel complexity than SVRG, and a better sequential complexity than the corresponding Stochastic Newton method (see Table 1).

## 2 Main result

In this section, we present our main result, which is the parallel and sequential complexity analysis of the local convergence of SVRN. The algorithm itself is discussed in detail in Section 3.

We now present the assumptions needed for our main result, starting with $\mu$-strong convexity of $f$ and $\lambda$-smoothness of each $\psi_i$. These are standard for establishing linear convergence rate of SVRG. Our result also requires Hessian regularity assumptions (Definition 1), which are standard for Newton's method and only affect the size of the local convergence neighborhood.

**Assumption 1** *We assume that $f(x) = \frac{1}{n}\sum_{i=1}^{n} \psi_i(x)$ has continuous first and second derivatives, as well as a bounded condition number $\kappa = \lambda/\mu$, where $\mu$ and $\lambda$ are defined as follows:*

    *1. Function $f$ is $\mu$-strongly convex, i.e.,*

$$f(\mathbf{x}) \geq f(\mathbf{x}') + \nabla f(\mathbf{x}')^\top(\mathbf{x} - \mathbf{x}') + \frac{\mu}{2}\|\mathbf{x} - \mathbf{x}'\|^2;$$

    *2. Each of the $n$ components $\psi_i$ is $\lambda$-smooth, i.e.,*

$$\psi_i(\mathbf{x}) \leq \psi_i(\mathbf{x}') + \nabla \psi_i(\mathbf{x}')^\top(\mathbf{x} - \mathbf{x}') + \frac{\lambda}{2}\|\mathbf{x} - \mathbf{x}'\|^2.$$

To highlight the parallelizability of SVRN due to large mini-batches, as well as the effect of variance reduction on its performance, we will consider two standard complexity measures:

    1. **Parallel complexity:** Number of batch gradient queries, i.e., times the algorithm computes a gradient at an iterate, over the full batch or a mini-batch. This corresponds to the PRAM model.

    2. **Sequential complexity:** Number of queries to component gradients $\nabla \psi_i(\mathbf{x})$, normalized by the total number of components $n$. Thus, one can think of this as the number of "data passes", where one "data pass" may possibly query component gradients at different iterates. This is a natural measure of complexity for stochastic first-order methods, which coincides with the parallel complexity when we only query full-batch gradients.

The motivation for this dual-complexity perspective is centered on the benefits of using large-batch gradient queries. These benefits come from the fact that on most computing architectures it takes far less time to compute a single gradient estimate on a batch of component functions at one location, than it takes to compute component gradients at different locations, one at a time. We highlight this by the notion of parallel complexity, which measures the number of batch gradient queries. Note that the benefits of large mini-batch sizes are not limited to parallel computing. Even standard single- and multi-core architectures benefit substantially from the vectorization of gradient computations, which is only effective when using large batches (as we observe in our experiments, see Section 5).

Despite these benefits, parallelization and vectorization still come with some computation/communication overhead, which is why it is natural to also optimize over the sequential complexity of the algorithms. Altogether, to express this in our model, we optimize both over parallel and sequential complexity, while prioritizing the parallel one. Note that for any algorithm using only full gradients (such as the standard versions of gradient descent or Subsampled Newton), the two notions of complexity are exactly equivalent. For example, in gradient descent (GD), both parallel and sequential complexity is $O(\kappa \log(1/\epsilon))$. By introducing stochastic gradients and variance reduction, as in SVRG, we can improve upon the sequential complexity of GD, while preserving (but not improving) its parallel complexity. Specifically, when $n \gg \kappa$, then SVRG with optimal mini-batch size takes $O(\log(1/\epsilon))$ sequential time to find an $\epsilon$-approximate solution, however it still needs $O(\kappa \log(1/\epsilon))$ parallel time (Table 1).

We can avoid the dependence of parallel complexity on the condition number $\kappa$ by using second-order information. In particular, suppose that in each iteration we compute the full gradient $\nabla f(\mathbf{x})$ and are given access to a (typically stochastic) Hessian estimate $\tilde{\mathbf{H}}$ such that for some $1 \leq \alpha \ll \kappa$,

$$\text{(Hessian } \alpha\text{-approximation)} \qquad \frac{1}{\sqrt{\alpha}} \nabla^2 f(\mathbf{x}) \preceq \tilde{\mathbf{H}} \preceq \sqrt{\alpha} \nabla^2 f(\mathbf{x}), \tag{2}$$

In fact, our condition can be stated in an even more general way, by asking that $\beta_1 \nabla^2 f(x) \preceq \tilde{\mathbf{H}} \preceq \beta_2 \nabla^2 f(x)$ for some fixed $0 < \beta_1 \leq \beta_2$ such that $\alpha = \beta_2/\beta_1$. In this case, $(1/\sqrt{\beta_1 \beta_2})\tilde{\mathbf{H}}$ is an $\alpha$-approximation in the sense of (2), which means that we can apply our results and algorithms given any such Hessian estimates.

A standard Stochastic Newton (SN) update, given below in (3), can achieve parallel and sequential complexity of $O(\alpha \log(1/\epsilon))$ locally in the neighborhood of the optimum. It is thus natural to ask whether we can use stochastic gradients and variance reduction to accelerate the local sequential complexity of this method, while preserving its parallel complexity. Our main result shows not only that this is possible, but also, remarkably, that this acceleration gets more significant the larger the data size $n$. See also Theorem 3 for algorithmic details and convergence analysis.

**Theorem 1 (informal Theorem 3)** *Suppose that Assumption 1 holds and: (a) $f$ has a Lipschitz Hessian, or (b) $f$ is self-concordant. Moreover, let $n \gg \kappa \gg \alpha$. There is an algorithm (SVRN) and an open neighborhood $U$ such that, given any $\mathbf{x} \in U$ with a corresponding Hessian $\alpha$-approximation as in (2), the cost of returning $\tilde{\mathbf{x}}$ such that $f(\tilde{\mathbf{x}}) - f(\mathbf{x}^*) \leq \epsilon \cdot (f(\mathbf{x}) - f(\mathbf{x}^*))$ is as follows:*

$$\text{Parallel time } = O\big(\alpha \log(1/\epsilon)\big) \text{ batch queries} \qquad and \qquad \text{Sequential time } = O\Big(\frac{\log(1/\epsilon)}{\log(n)}\Big) \text{ data passes.}$$

**Remark 1** *SVRN improves on the sequential complexity of Stochastic Newton by $O(\alpha \log(n))$, while retaining the same parallel complexity. Moreover, if $\alpha \leq 2$, then the algorithm accepts a unit step size, and still achieves $O(\log(n))$ acceleration. Note that this acceleration improves with the problem size $n$, which is a unique property of SVRN.*

**Remark 2** *To find an initialization point for SVRN, one can simply run a few iterations of a Subsampled Newton method with line search. In Section 4, we propose a globally convergent algorithm based on this approach (SVRN-HA; see Algorithm 1), and in Section 5 we show empirically that it substantially accelerates Subsampled Newton.*

|  | Second-order | Parallel (batch queries) | Sequential (data passes) |
|---|:---:|:---:|:---:|
| Gradient Descent | ✗ | $O(\kappa \log(1/\epsilon))$ | $O(\kappa \log(1/\epsilon))$ |
| Accelerated GD | ✗ | $O(\sqrt{\kappa} \log(1/\epsilon))$ | $O(\sqrt{\kappa} \log(1/\epsilon))$ |
| Stochastic Newton | ✓ | $O(\alpha \log(1/\epsilon))$ | $O(\alpha \log(1/\epsilon))$ |
| Mini-batch SVRG | ✗ | $O(\kappa \log(1/\epsilon))$ | $O(\log(1/\epsilon))$ |
| Mini-batch Katyusha | ✗ | $O(\sqrt{\kappa} \log(1/\epsilon))$ | $O(\log(1/\epsilon))$ |
| **SVRN (this work)** | ✓ | $O(\alpha \log(1/\epsilon))$ | $O\big(\frac{\log(1/\epsilon)}{\log(n)}\big)$ |

Table 1: Comparison of local convergence behavior for SVRN and related stochastic methods in the big data regime, i.e., $n \gg \kappa$, along with full-batch Gradient Descent (GD), and Accelerated GD. Time complexities are obtained by first optimizing parallel time (batch queries), and then optimizing sequential time (data passes). For the second-order methods, we assume a Hessian $\alpha$-approximation (2) where $\alpha \ll \kappa$.

## 2.1 Discussion

In this section, we compare the local convergence complexity of SVRN to standard stochastic first-order and second-order algorithms. In this comparison, we focus on what we call the big data regime, i.e., $n \gg \kappa$, which is of primary interest in the literature on Subsampled Newton methods. Then, in Section 2.2, we illustrate how SVRN can be further improved via sketching and importance sampling, when solving problems with additional structure, such as least squares.

**Comparison to SVRG and Katyusha.** As we can see in Table 1, first-order algorithms, including variance-reduced methods such as SVRG (Johnson & Zhang, 2013), and its accelerated variants like Katyusha (Allen-Zhu, 2017), suffer from a dependence on the condition number $\kappa$ in their parallel complexity. Namely, they require either $O(\kappa \log(1/\epsilon))$ or $O(\sqrt{\kappa} \log(1/\epsilon))$ batch gradient queries, compared to $O(\alpha \log(1/\epsilon))$ for SVRN and Stochastic Newton, where $\alpha$ is the Hessian approximation factor, which is often much smaller than $\sqrt{\kappa}$. This is because, unlike SVRN, these methods do not scale well to large mini-batches, making them less parallelizable.

Another difference between SVRN and SVRG or Katyusha is that, when the Hessian approximation is sufficiently accurate ($\alpha \leq 2$), then SVRN accepts a unit step size, which leads to optimal convergence rate without any tuning. On the other hand, the optimal step size for SVRG depends on the strong convexity and smoothness constants $\mu$ and $\lambda$, and thus, requires tuning.

**Comparison to Stochastic Newton.** We next compare SVRN with Stochastic Newton methods such as Subsampled Newton and Newton Sketch. Here, the most standard proto-algorithm considered in the literature is the following update:

$$\tilde{\mathbf{x}}_{s+1} = \tilde{\mathbf{x}}_s - \eta_s \tilde{\mathbf{H}}^{-1} \nabla f(\tilde{\mathbf{x}}_s). \tag{3}$$

As mentioned earlier, this update uses only full gradients, so both its parallel and sequential complexity is $O(\alpha \log(1/\epsilon))$ (see Stochastic Newton in Table 1). On the other hand, SVRN provides a direct acceleration of the sequential complexity without sacrificing any parallel complexity.

The Hessian $\alpha$-approximation $\tilde{\mathbf{H}} \approx \nabla^2 f(\tilde{\mathbf{x}}_s)$ can be produced in a number of ways, but perhaps the most relevant for this discussion is Hessian subsampling, a.k.a. Subsampled Newton (e.g., see Roosta-Khorasani & Mahoney, 2019). In this setting, given $k$ component Hessians sampled uniformly at random, we can with high probability construct an estimate $\tilde{\mathbf{H}}$ with approximation factor $\alpha = 1 + O\big(\kappa \log(d)/k + \sqrt{\kappa \log(d)/k}\big)$ (see Appendix D.2), so that $\alpha \ll \kappa$ for any $k \gg \log(d)$, and $\alpha \leq 2$ for $k = O(\kappa \log(d))$. However, if we wanted to recover SVRN's sequential complexity of $O\big(\frac{\log(1/\epsilon)}{\log(n)}\big)$ purely by improving the Hessian approximation in Subsampled Newton, the required Hessian sample size $k$ would become at least as large as $n$, meaning that we would essentially have to use the exact Hessian (i.e., Newton's method), which is highly undesirable.

We note that some of the literature on Subsampled Newton proposes to subsample both the Hessian and the gradient (e.g., Bollapragada et al., 2018), which would be akin to $\tilde{\mathbf{x}}_{s+1} = \tilde{\mathbf{x}}_s - \eta_s \tilde{\mathbf{H}}^{-1} \frac{1}{m} \sum_{i=1}^{m} \nabla \psi_{I_i}(\tilde{\mathbf{x}}_s)$. However, as is noted in the literature, to maintain linear convergence of such a method, one has to keep increasing the gradient sample size at an exponential rate, which means that, for finite-sum minimization, we quickly revert back to the full gradient (see experiments in Section 5.2).

## 2.2 Accelerating SVRN with sketching and importance sampling

When the minimization task possesses additional structure, then we can combine SVRN with Hessian and gradient estimation techniques other than uniform subsampling. For example, one such family of techniques, called randomized sketching (Drineas & Mahoney, 2016; Woodruff, 2014; Murray et al., 2023; Dereziński & Mahoney, 2024), is applicable when the Hessian can be represented by a decomposition $\nabla^2 f(\mathbf{x}) = \mathbf{A}_f(\mathbf{x})^\top \mathbf{A}_f(\mathbf{x}) + \mathbf{C}$, where $\mathbf{A}_f(\mathbf{x})$ is a tall $n \times d$ matrix and $\mathbf{C}$ is a fixed $d \times d$ matrix. This setting applies for many empirical risk minimization tasks, including linear and logistic regression, among others.

Sketching can be used to construct an estimate of the Hessian by applying a randomized linear transformation to $\mathbf{A}_f(\mathbf{x})$, represented by a $k \times n$ random matrix $\mathbf{S}$, where $k = \tilde{O}(d)$ is much smaller than $n$. Using standard sketching techniques, such as Subsampled Randomized Hadamard Transforms (SRHT, Ailon & Chazelle, 2009), Sparse Johnson-Lindenstrauss Transforms (SJLT, Clarkson & Woodruff, 2017; Nelson & Nguyên, 2013; Chenakkod et al., 2024) and Leverage Score Sparsified embeddings (LESS, Dereziński et al., 2021), we can construct a Hessian estimate that satisfies the requirements of Theorem 1 at the cost of $\tilde{O}(nd + d^3)$, which corresponds to a nearly-constant number of data passes and $d \times d$ matrix multiplies. In particular, this eliminates the dependence of Hessian estimation on the condition number.

Another way of making SVRN more efficient is to use importance sampling in the stochastic gradient mini-batches. Importance sampling can be introduced to any finite-sum minimization task (1) by specifying an $n$-dimensional probability vector $p = (p_1, ..., p_n)$, such that $\sum_i p_i = 1$, and sampling the component gradient $\psi_{I_i}(\mathbf{x})$ so that the index $I_i$ is drawn according to $p$. With the right choice of importance sampling, we can substantially reduce the smoothness parameter $\lambda$, and thereby, the condition number $\kappa$ of the problem (see Appendix A.4).

The above techniques can be used to accelerate SVRN, for instance, in the important task of solving least squares regression. Here, given an $n \times d$ matrix $\mathbf{A}$ with rows $\mathbf{a}_i^\top$ and an $n$-dimensional vector $\mathbf{y}$, the objective being minimized is the following quadratic function:

$$f(\mathbf{x}) = \frac{1}{2n} \|\mathbf{A}\mathbf{x} - \mathbf{y}\|^2 = \frac{1}{n} \sum_{i=1}^{n} \frac{1}{2} (\mathbf{a}_i^\top \mathbf{x} - y_i)^2. \tag{4}$$

One of the popular methods for solving the least squares task, known as the Iterative Hessian Sketch (IHS, Pilanci & Wainwright, 2016), is exactly the Stochastic Newton update (3), where the Hessian estimate $\tilde{\mathbf{H}}$ is constructed via sketching. In this context, SVRN can be viewed as an accelerated version of IHS. To fully leverage the structure of the least squares problem, we use a popular importance sampling technique called leverage score sampling (Drineas et al., 2006; 2012), where the importance probabilities are (approximately) proportional to $p_i \propto \mathbf{a}_i^\top (\mathbf{A}^\top \mathbf{A})^{-1} \mathbf{a}_i$. Through an adaptation of our main result, we show that a version of SVRN for least squares, with sketched Hessian and leverage score sampled gradients, improves on the state-of-the-art complexity for reaching a high-precision solution to a preconditioned least squares task from $O(nd \log(1/\epsilon))$ (Rokhlin & Tygert, 2008; Avron et al., 2010; Meng et al., 2014) to $O\left(nd \frac{\log(1/\epsilon)}{\log(n/d)}\right)$. See Appendix A.4 for proof and further discussion.

**Theorem 2 (Fast least squares solver)** *Given $\mathbf{A} \in \mathbb{R}^{n \times d}$ and $\mathbf{y} \in \mathbb{R}^n$, after $O(nd \log n + d^3 \log d)$ pre-processing cost to find the sketched Hessian estimate $\tilde{\mathbf{H}}$ and an approximate leverage score distribution, SVRN finds $\tilde{\mathbf{x}}$ so that*

$$f(\tilde{\mathbf{x}}) \le (1 + \epsilon) f(\mathbf{x}^*) \quad in \quad O\left(nd \frac{\log(1/\epsilon)}{\log(n/d)}\right) \quad time.$$

Crucially, the SVRN-based least squares solver only requires a preconditioner $\tilde{\mathbf{H}}$ that is a constant factor approximation of the Hessian, i.e., $\alpha = O(1)$. Interestingly, our approach of transforming the problem via leverage score sampling appears to be connected to a weighted and preconditioned SGD algorithm of (Yang et al., 2017) for solving a more general class of $\ell_p$-regression problems. We expect that Theorem 2 can be similarly extended beyond least squares regression.

### 2.3 Further related work

As mentioned in Section 1, a number of works have aimed to accelerate first-order variance reduction methods by preconditioning them with second-order information. For example, (Gonen et al., 2016) proposed Preconditioned SVRG for ridge regression. The effect of this preconditioning, as in related works (Liu et al., 2019), is a reduced condition number $\kappa$ of the problem. This is different from Theorem 1, which uses preconditioning to make variance reduction effective for large mini-batches and with a unit step size.

Some works have shown that, instead of preconditioning, one can use momentum to accelerate variance reduction, and also to improve its convergence rate when using mini-batches. These methods include Catalyst (Lin et al., 2015) and Katyusha (Allen-Zhu, 2017). However, unlike SVRN, these approaches are still limited to fairly small mini-batches, as demonstrated in Table 1.

A number of works have proposed applying techniques inspired by variance reduction to stochastic Newton-type methods in settings which are largely incomparable to ours. First, (Rodomanov & Kropotov, 2016; Kovalev et al., 2019) consider algorithms where the Hessian and gradient information is incrementally updated with either individual samples or mini-batches. However, the approximate Hessian information required by these methods is quite different than the one used in SVRN: they require the Hessian estimate to be initialized with *all* $n$ component Hessians, possibly computed at different locations (compared to, e.g., a subsampled estimate). For example, in the case of least squares, this means computing the exact Hessian, which costs $O(nd^2)$ time and renders the task trivial (compare this to our Theorem 2, where the Hessian estimate required by SVRN can be approximated efficiently). Setting this aside, we can still compare the local convergence rate of SVRN with the Stochastic Newton method of (Kovalev et al., 2019, Theorem 1) using the same mini-batch size. Assuming $n \gg \kappa$ and using the setup from Theorem 1, their method achieves $O(\log(1/\epsilon))$ data passes, whereas SVRN obtains the accelerated complexity of $O\big(\frac{\log(1/\epsilon)}{\log(n)}\big)$ data passes.

In the non-convex setting, variance reduction was used by (Zhou et al., 2019; Zhang et al., 2022) to accelerate Subsampled Newton with cubic regularization. They use variance reduction both for the gradient and the Hessian estimates. Also, (Wang et al., 2017) incorporate variance reduction into a stochastic quasi-Newton method. However, due to the non-convex setting, these results are incomparable to ours, as we are focusing on strongly convex optimization.

## 3 Local convergence analysis of SVRN

In this section, we present the convergence analysis for SVRN, leading to the proof of Theorem 1.

**Notation.** For $d \times d$ positive semidefinite matrices $\mathbf{A}$ and $\mathbf{B}$, we define $\|\mathbf{v}\|_{\mathbf{A}} = \sqrt{\mathbf{v}^\top \mathbf{A} \mathbf{v}}$, and we say that $\mathbf{A} \approx_\epsilon \mathbf{B}$, when $(1 - \epsilon)\mathbf{B} \preceq \mathbf{A} \preceq (1 + \epsilon)\mathbf{B}$, where $\preceq$ denotes the positive semidefinite ordering (we define analogous notation $a \approx_\epsilon b$ for non-negative scalars $a, b$). We use $c$ and $C$ to denote positive absolute constants, and let $I \sim [n]$ denote a uniformly random sample from $\{1, .., n\}$.

We will present the analysis in a slightly more general setting of expected risk minimization, i.e., where $f(\mathbf{x}) = \mathbb{E}_{\psi \sim \mathcal{D}}[\psi(\mathbf{x})]$. Here, $\mathcal{D}$ is a distribution over convex functions $\psi : \mathbb{R}^d \to \mathbb{R}$. Clearly, this setting subsumes (1), since we can let $\mathcal{D}$ be a uniformly random sample $\psi_i$. Thanks to this extension, our results can apply to *importance* sampling of component functions, as in Theorem 2.

**Definition 1** *We define the local convergence neighborhood $U_f(\epsilon)$, parameterized by $\epsilon \in (0, 1)$, as:*

1. *If $f$ has an $L$-Lipschitz Hessian, then $U_f(\epsilon) = \{\mathbf{x} : \|\mathbf{x} - \mathbf{x}^*\|_{\nabla^2 f(\mathbf{x}^*)} < \epsilon\, \mu^{3/2}/L\}$;*

2. *If $f$ is self-concordant, then we use $U_f(\epsilon) = \{\mathbf{x} : \|\mathbf{x} - \mathbf{x}^*\|_{\nabla^2 f(\mathbf{x}^*)} < \epsilon/4\}$.*

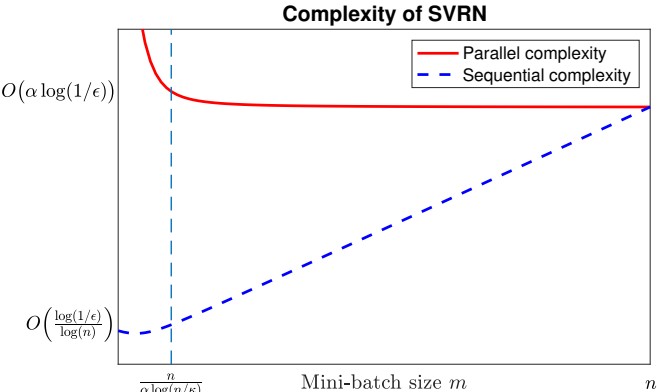

Figure 1: Illustration of the local convergence complexity analysis for SVRN, as a function of the mini-batch size $m$, with the number of inner iterations set to $t_{\max} = n/m$. As we decrease the mini-batch size from $n$ (standard Stochastic Newton; SN) downto $m \approx \frac{n}{\alpha \log(n/\kappa)}$ (optimal SVRN), the sequential complexity (number of passes over the data) improves by $O(\alpha \log(n))$, while the parallel complexity (number of batch gradient queries) remains optimal.

Our local convergence analysis is captured by the following theorem, which provides the rate of convergence after one outer iteration of SVRN (stated below), for a range of mini-batch sizes $m$.

**Theorem 3 (Convergence rate of SVRN)** *Suppose that Assumption 1 holds, $\alpha \geq 1$, and either: (a) $f$ has a Lipschitz continuous Hessian, or (b) $f$ is self-concordant. There is an absolute constant $c > 0$ such that if $\tilde{\mathbf{x}}_s \in U_f(1/c\alpha)$, and we are given the gradient $\tilde{\mathbf{g}}_s = \nabla f(\tilde{\mathbf{x}}_s)$ as well as a Hessian $\alpha$-approximation, i.e., $\tilde{\mathbf{H}}$ such that $\frac{1}{\sqrt{\alpha}} \nabla^2 f(\tilde{\mathbf{x}}_s) \preceq \tilde{\mathbf{H}} \preceq \sqrt{\alpha} \nabla^2 f(\tilde{\mathbf{x}}_s)$, then, letting $\mathbf{x}_0 = \tilde{\mathbf{x}}_s$ and:*

$$\mathbf{x}_{t+1} = \mathbf{x}_t - \eta \tilde{\mathbf{H}}^{-1} \left( \frac{1}{m} \sum_{i=1}^{m} \nabla \psi_i(\mathbf{x}_t) - \nabla \psi_i(\tilde{\mathbf{x}}_s) + \tilde{\mathbf{g}}_s \right), \qquad \psi_1, ..., \psi_m \sim \mathcal{D},$$

*after $t$ iterations with mini-batch size $m \geq c\alpha^2 \kappa \log(t/\delta)$ and step size $\eta = \min\{\sqrt{2/\alpha}, 1\}$, the iterate $\tilde{\mathbf{x}}_{s+1} = \mathbf{x}_t$ (i.e., one outer iteration of SVRN) with probability $1 - \delta$ satisfies:*

$$\frac{f(\tilde{\mathbf{x}}_{s+1}) - f(\mathbf{x}^*)}{f(\tilde{\mathbf{x}}_s) - f(\mathbf{x}^*)} \leq \left( 1 - \frac{1}{2\alpha} \right)^t + c\alpha^2 \log(t/\delta) \frac{\kappa}{m}.$$

**Remark 3** *The dependence on the condition number $\kappa$ in our result comes from sub-sampled gradient estimation. This is consistent with Subsampled Newton works such as Roosta-Khorasani & Mahoney (2019): to recover their fast condition number-free convergence guarantees they require the gradient sample size to be sufficiently larger than $\kappa$. We showed how this can be avoided for certain losses (least squares; see Theorem 2 and Lemma 6) by relying on importance sampling.*

The proof of Theorem 3, which is given in Appendix A, relies on a new high-probability bound for the error of the variance-reduced gradient estimates in the large mini-batch regime, measured using the vector norm defined by the inverse Hessian at the optimum (Lemma 3). Unlike results from prior work, which hold in expectation, this bound crucially relies on the iterate being in the local neighborhood. Also, unlike standard SVRG analysis, we achieve our convergence guarantee for the *last iterate* of SVRN's inner loop (as opposed a random or averaged iterate), which is again enabled by exploiting local second-order information.

**Discussion.** For simplicity, let us fix the number of inner iterations $t_{\max} = n/m$, so that a single outer iteration of SVRN always takes two passes over the data. Then, we can define the linear convergence rate after one outer iteration as a function of mini-batch size $m$:

$$\rho_m := \left( 1 - \frac{1}{2\alpha} \right)^{n/m} + \tilde{O}(\kappa/m).$$

**Input**: iterate $\tilde{\mathbf{x}}_0$, gradient batch size $m$, Hessian sample size $k$, and local iterations $t_{\max}$;
Initialize step size $\eta_{-1} = 0$ and Hessian estimate $\tilde{\mathbf{H}}_{-1} = \mathbf{0}$;
**for** $s = 0, 1, 2, \ldots$ **do**
    Compute the subsampled Hessian: $\widehat{\mathbf{H}}_s = \frac{1}{k} \sum_{i=1}^{k} \nabla^2 \psi_i(\tilde{\mathbf{x}}_s)$,    for    $\psi_1, ..., \psi_k \sim \mathcal{D}$;
    Compute the Hessian average: $\tilde{\mathbf{H}}_s = \frac{s}{s+1} \tilde{\mathbf{H}}_{s-1} + \frac{1}{s+1} \widehat{\mathbf{H}}_s$;
    Compute the full gradient: $\tilde{\mathbf{g}}_s = \nabla f(\mathbf{x}_s)$;
    **if** $\eta_{s-1} < 1$ **then**
        Compute the descent direction $\tilde{\mathbf{v}}_s$ by solving: $\tilde{\mathbf{H}}_s \tilde{\mathbf{v}}_s = -\tilde{\mathbf{g}}_s$;
    **else**
        Initialize $\mathbf{x}_0 = \tilde{\mathbf{x}}_s$;
        **for** $t = 0, \ldots, t_{\max} - 1$ **do**
            Compute $\widehat{\mathbf{g}}_t(\mathbf{x}_t)$ and $\widehat{\mathbf{g}}_t(\tilde{\mathbf{x}}_s)$,    for    $\widehat{\mathbf{g}}_t(\mathbf{x}) = \frac{1}{m} \sum_{i=1}^{m} \nabla \psi_i(\mathbf{x})$,    $\psi_1, ..., \psi_m \sim \mathcal{D}$;
            Compute variance-reduced gradient $\bar{\mathbf{g}}_t = \widehat{\mathbf{g}}_t(\mathbf{x}_t) - \widehat{\mathbf{g}}_t(\tilde{\mathbf{x}}_s) + \tilde{\mathbf{g}}_s$;
            Compute the descent direction $\mathbf{v}_t$ by solving: $\tilde{\mathbf{H}}_s \mathbf{v}_t = -\bar{\mathbf{g}}_t$;
            Update $\mathbf{x}_{t+1} = \mathbf{x}_t + \mathbf{v}_t$
        **end**
        Compute the descent direction: $\tilde{\mathbf{v}}_s = \mathbf{x}_{t_{\max}} - \tilde{\mathbf{x}}_s$;
    **end**
    Compute $\eta_s$ for iterate $\tilde{\mathbf{x}}_s$ and direction $\tilde{\mathbf{v}}_s$ using the Armijo condition;
    Update $\tilde{\mathbf{x}}_{s+1} = \tilde{\mathbf{x}}_s + \eta_s \tilde{\mathbf{v}}_s$;
**end**



**Algorithm 1:** SVRN with Hessian Averaging (SVRN-HA)



Let us assume the big data regime, i.e., $n \gg \kappa$. If we only use full-batch gradients ($m = n$), then the first term in the rate dominates, and we have $\rho_m \approx 1 - \frac{1}{2\alpha}$, which is similar to what we would get using standard Stochastic Newton (3). As we decrease $m$ (and change $t_{\max}$ accordingly), the first term in $\rho_m$ decreases, whereas the second term increases. As a result, the overall rate rapidly improves, reaching its optimal value of $\rho_m = \tilde{O}(\kappa/n)$ for $m \approx \frac{n}{\alpha \log(n/\kappa)}$.

**Complexity analysis.** The complexity analysis given in Theorem 1 follows directly from the above discussion, since the sequential complexity (number of data passes needed to improve by factor $\epsilon$) is given by $O\left(\frac{\log(1/\epsilon)}{\log(1/\rho_m)}\right)$, whereas the parallel complexity (number of batch gradient queries) is $O\left(t_{\max} \cdot \frac{\log(1/\epsilon)}{\log(1/\rho_m)}\right)$. In Figure 1, we illustrate how these quantities change as a function of $m$. In particular, we observe that the batch gradients essentially stay flat at $O(\alpha \log(1/\epsilon))$ as we decrease $m$, until reaching $\frac{n}{\alpha \log(n/\kappa)}$. On the other hand, the data pass complexity decreases linearly with $m$, until it reaches the optimal value of $O\left(\frac{\log(1/\epsilon)}{\log(n/\kappa)}\right)$, which, for sufficiently large $n$, recovers Theorem 1.

# 4 Globally convergent algorithm

We next present a practical stochastic second-order method (see Algorithm 1, called SVRN-HA) which uses SVRN to accelerate its local convergence phase.

The key in implementing SVRN is that the algorithm is guaranteed to converge with unit step size only once we reach a local neighborhood of the optimum, and if we have a sufficiently accurate Hessian estimate. For this reason, we introduce an initial phase of the algorithm, in which a standard Stochastic Newton method is ran, using the Armijo line search to select the step size. Once the method reaches the local convergence neighborhood, as long as the Hessian estimates are accurate enough, the line search is guaranteed to return a unit step size. At this point, the algorithm switches to SVRN and achieves acceleration. Finally, to ensure that we reach a sufficiently accurate Hessian estimate, our Stochastic Newton method should gradually increase the accuracy of the Hessian estimates.

Based on these insights, we propose an algorithm called Stochastic Variance-Reduced Newton with Hessian Averaging (SVRN-HA). In the initial phase, this algorithm is a variant of Subsampled Newton, based on a method proposed by (Na et al., 2022), where, at each iteration, we construct a subsampled Hessian estimate based on a fixed sample size $k$. To increase the accuracy over time, all past Hessian estimates are averaged together, and the result is used to precondition the full gradient. At each iteration, we check whether the last line search returned a unit step size. If yes, then we start running SVRN with local iterations $t_{\max} = \lfloor \log_2(n/d) \rfloor$ and gradient batch size $m = \lfloor n/\log_2(n/d) \rfloor$, where $n$ is the number of data points and $d$ is the dimension. This is motivated by our theory (see discussion below Theorem 3), using $d$ as a proxy for the condition number $\kappa$. This choice has proven effective in all of the evaluated datasets, which indicates that the theoretical dependence of the mini-batch size on the condition number is likely quite pessimistic.

In the following result, we establish global convergence of SVRN-HA, by showing that the global phase of this method will not only reach any local neighborhood, but also that the Hessian estimate will get progressively more accurate, eventually reaching the desired approximation accuracy.

**Theorem 4** *Let $f$ be as in Theorem 3. For any neighborhood $U$ around the optimum, Algorithm 1 will almost surely reach a point where: (a) $\tilde{\mathbf{x}}_s$ belongs to the neighborhood $U$, and (b) the Hessian estimate $\tilde{\mathbf{H}}_s$ satisfies the condition in Theorem 3. At this point, the line search will return $\eta_s = 1$.*

## 5 Experiments

We next demonstrate numerically that SVRN can be effectively used to accelerate stochastic Newton methods in practice. We also show how variance reduction can be incorporated into a globally convergent Subsampled Newton method in a way that is robust to hyperparameters and preserves its scalability thanks to large-batch operations.[1]

### 5.1 Logistic regression experiment

In this section, we present numerical experiments for solving a regularized logistic loss minimization task. For an $n \times d$ data matrix $\mathbf{A}$ with rows $\mathbf{a}_i^\top$, an $n$-dimensional target vector $\mathbf{y}$ (with $\pm 1$ entries $y_i$) and a regularization parameter $\gamma$, our task is to minimize:

$$f(\mathbf{x}) = \frac{1}{n} \sum_{i=1}^{n} \log(1 + \mathrm{e}^{-y_i \mathbf{a}_i^\top \mathbf{x}}) + \frac{\gamma}{2}\|\mathbf{x}\|^2. \tag{5}$$

As a dataset, we used the Extended MNIST dataset of handwritten digits (EMNIST, Cohen et al., 2017) with $n = 500$k datapoints, transformed using a random features map (with dimension $d = 1000$). Experimental details, as well as further results on the CIFAR-10 dataset and several synthetic data matrices, are presented in Appendix B.

In Figure 2, we compared SVRN-HA to three baselines which are most directly comparable: (1) the classical Newton's method; (2) SVRG with the step size and number of inner iterations tuned for best wall-clock time; and (3) Subsampled Newton with Hessian Averaging (SN-HA), i.e., the method we use in the global phase of Algorithm 1 (without the SVRN phase). All of the convergence plots are averaged over 10 runs. For both SVRN-HA and SN-HA we use Hessian sample size $k = 4d$.

From Figure 2(a), we conclude that as soon as SVRN-HA exits the initial phase of the optimization, it accelerates dramatically, to the point where it nearly matches the rate of classical Newton. This acceleration corresponds to the improvement in sequential complexity from $O(\alpha \log(1/\epsilon))$ for Stochastic Newton to $O(\frac{\log(1/\epsilon)}{\log(n)})$ for SVRN. In all our experiments, the transition to the SVRN phase in SVRN-HA occurred very quickly, generally within 1-2 iterations, which indicates that the method easily reaches local convergence. Finally, we observed that the convergence of SVRG is initially quite fast, but over time, it stabilizes at a slower rate, indicating that the Hessian information plays a significant role in the performance of SVRN-HA.

---

[1]The code is available at https://github.com/svrnewton/svrn.

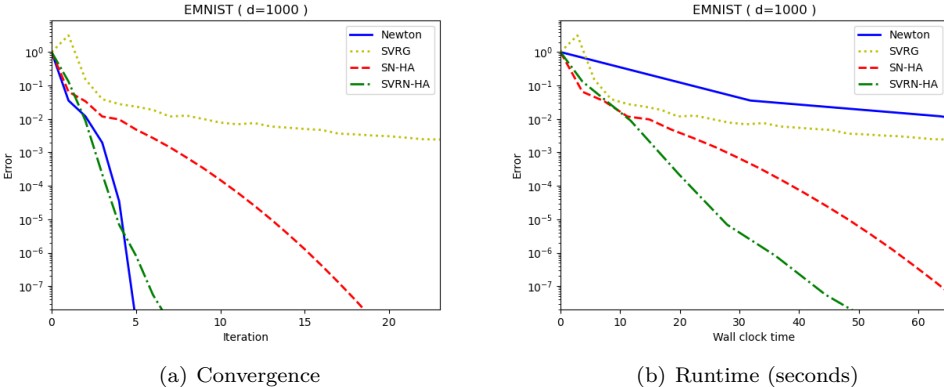

(a) Convergence           (b) Runtime (seconds)

Figure 2: Convergence and runtime comparison of SVRN-HA on the EMNIST dataset against three baselines: classical Newton, SVRG (after parameter tuning), and Subsampled Newton with Hessian Averaging (SN-HA), i.e., the global phase of Algorithm 1, ran without switching to SVRN. Further results on the CIFAR-10 dataset are in Appendix B.

In Figure 2(b), we plot the wall clock time of the algorithms. Here, SVRN-HA also performs better than all of the baselines, despite some additional per-iteration overhead. We expect that this can be further optimized. Finally, we note that Newton's method is drastically slower than all other methods due to the high cost of solving a large linear system, and the per-iteration time of SVRG is substantially slowed by its sequential nature.

## 5.2 Further investigations on a least squares task

We next study the setting of least squares regression (4) to analyze the trade-offs in convergence for different implementations of SVRN, as we vary the gradient and Hessian estimation schemes. We evaluated the algorithms on synthetic data matrices, as defined in Appendix B.

**Communication cost of gradient resampling.** Our theoretical analysis requires that for each small step of SVRN, a fresh sample of components $\psi_i$ is used to compute the gradient estimates. However, in Lemma 3 we showed that, after variance reduction, the gradient estimates are accurate with high probability, which suggests that we might be able to reuse previously sampled components. While this technically does not improve the number of required gradient queries, it can substantially reduce the communication cost for some practical implementations.

As an example, let us consider the setting where each full/mini-batch gradient computation requires reading the corresponding data chunk from the server onto the computing core. Then, the theoretical version of SVRN requires reading the entire dataset roughly 2 times ($2n$ data points) in the course of one stage (outer iteration): $n$ data points for computing the full gradient, and then $t_{\max} \cdot m = (n/m) \cdot m = n$ data points for all of the mini-batch steps together. On the other hand, if we were to reuse the same mini-batch for all of the inner iterations in one stage, then we require reading the dataset only $1 + o(1)$ times.

In Figure 3(a), we investigate how much the convergence rate of SVRN-HA is affected by the frequency of component resampling for the gradient estimates. Recall that in all our experiments, we use a gradient sample size of $m = \lfloor n/\log_2(n/d) \rfloor$. We consider the following variants:

1. Sampling once: an extreme policy of sampling one set of components and reusing them for all gradient estimates;

2. Sampling per stage: an intermediate policy of resampling the components after every full gradient computation.

3. Sampling per step: the policy which is used in our theory, i.e., resampling the gradients at every step of the inner loop of the algorithm.

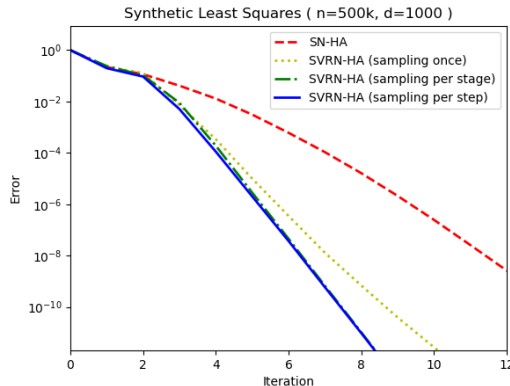 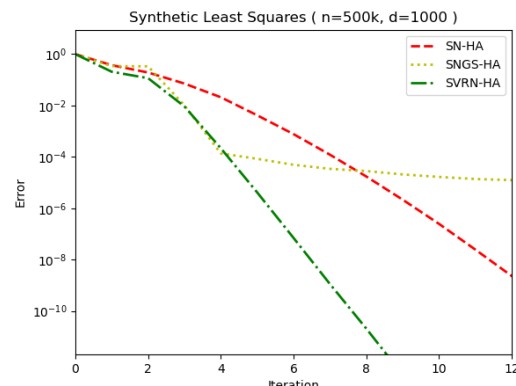

(a) Comparison of three variants of SVRN-HA (alongside SN-HA), depending on how frequently we resample data points used to compute the gradient estimate. We consider three variants of SVRN-HA: (1) sampling once for the entire optimization, (2) sampling once for each full gradient stage (per stage), (3) sampling in each small step (per step).

(b) Comparison of SVRN-HA (alongside SN-HA) against Subsampled Newton with Gradient Subsampling (SNGS-HA), which is implemented exactly like SVRN-HA except without the variance-reducing correction.

Figure 3: How different types of gradient estimation affect the convergence properties of SVRN.

From Figure 3(a) we conclude that, while all three variants of SVRN-HA converge and are competitive with SN-HA, the extreme policy of sampling once leads to a substantial degradation in convergence rate, whereas sampling per stage and sampling per step perform very similarly. Thus, our overall recommendation is to resample the components at every stage of SVRN-HA, but reuse the sample for the small steps of the algorithm (this is what we used for the EMNIST and CIFAR-10 experiments).

**Effect of variance reduction.** We next investigate the effect of variance reduction on the convergence rate of SVRN. While gradient subsampling has been proposed by many works in the literature on Subsampled Newton (e.g., see Roosta-Khorasani & Mahoney, 2019), these works have shown that the gradient sample size must be gradually increased to retain fast local convergence (which means that after a few iterations, we must use the full gradient). On the other hand, in SVRN, instead of increasing the gradient sample size, we use variance reduction with a fixed sample size, which allows us to retain the accelerated convergence indefinitely.

To illustrate this point, in Figure 3(b) we plot how the convergence behavior of our algorithm changes if we take variance reduction out of it. The resulting method is called Subsampled Newton with Gradient Subsampling (SNGS-HA). For this experiment, we resample the gradient estimate at every small step (for both SNGS-HA and SVRN-HA). For the sake of direct comparison, all of the other parameters are retained from SVRN-HA. In particular, one iteration of SNGS-HA corresponds to $\lfloor \log_2(n/d) \rfloor$ steps using resampled gradients, and Hessian averaging occurs once every such iteration. As expected, we observe that, while initially converging at a fast rate, eventually SNGS-HA reaches a point where the subsampled gradient estimates are not sufficiently accurate, resulting in a sudden dramatic drop-off in the convergence rate, to the point where the method virtually stops converging altogether. On the other hand, SVRN-HA continues to converge at the same fast rate throughout the optimization procedure without any reduction in performance. This indicates that variance reduction does improve the accuracy of gradient estimates, especially when our goal is to converge to a high-precision solution.

**Effect of Hessian accuracy.** In our experiments, for both SVRN and SN, we used Hessian averaging (Na et al., 2022) to construct the Hessian estimates. This approach is desirable in practice, since it gradually increases the accuracy of the Hessian estimate as we progress in the optimization. As a result, it is more robust to the Hessian sample size and we are guaranteed to reach sufficient accuracy for SVRN to work well. In the following experiment, we take Hessian averaging out of the algorithms to provide a better sense of how

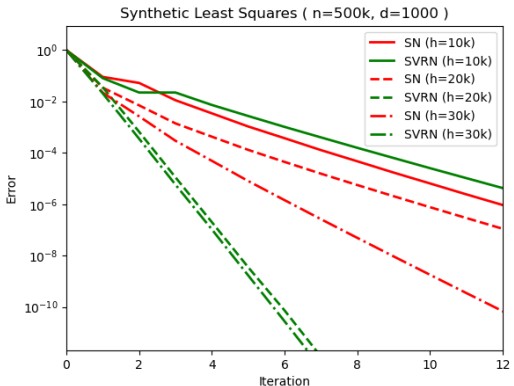
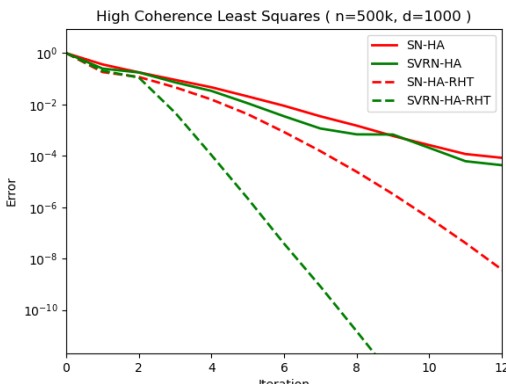

(a) Convergence comparison of SVRN and SN using fixed Hessian estimates (i.e., without Hessian averaging). Here, $h$ denotes the number of Hessian samples used to generate the estimate.

(b) Convergence comparison of SVRN-HA and SN-HA, with and without preconditioning using a Randomized Hadamard Transform (RHT), for a high-coherence least squares dataset.

Figure 4: How Hessian sample size and data coherence affect the convergence properties of SVRN.

the performance of SVRN and SN depends on the accuracy of the provided Hessian estimate. For simplicity, we focus here on least squares, where the Hessian is the same everywhere, so we can simply construct an initial Hessian estimate and then use it throughout the optimization. However, our insights apply more broadly to local convergence for general convex objectives. In Figure 4(a), we plot the performance of the algorithms as we vary the accuracy of the subsampled Hessian estimates, where $h$ denotes the number of samples used to construct the estimate. In all the results, we keep the gradient sample size and local steps in SVRN fixed as before.

Remarkably, the performance of SVRN is affected by the Hessian accuracy very differently than SN. We observe that SVRN requires a certain level of Hessian accuracy to provide any acceleration over SN. As soon as this level of Hessian accuracy is reached (by increasing the Hessian sample size $h$), the peformance of SVRN quickly jumps to the fast convergence we observed in the other experiments. Further increasing the accuracy no longer appears to affect the convergence rate of SVRN. This is in contrast to SN, whose convergence slowly improves as we increase the Hessian sample size. This intriguing phenomenon is actually fully predicted by our theory for SVRN (together with prior convergence analysis for SN). Our convergence result (Theorem 3) requires a sufficiently accurate Hessian inverse estimate for SVRN to work with a unit step size (which is what is used in SVRN-HA), but the actual rate of convergence is independent of the Hessian accuracy (only the required number of small steps is affected). We conclude that SVRN is more desirable than SN when we have a small budget for Hessian samples.

**Effect of high coherence.** We next analyze the performance of SVRN and SN on a slightly modified least squares task. For this experiment, we modify the data matrix $\mathbf{A}$, by multiplying the $i$th row by $1/\sqrt{g_i}$ for each $i$, where $g_i$ is an independent random variable distributed according to the Gamma distrution with shape 2 and scale $1/2$. This is a standard transformation designed to produce a matrix with many rows having a high leverage score. Recall that the leverage score of the $i$th row of $\mathbf{A}$ is defined as $\ell_i = \mathbf{a}_i^\top (\mathbf{A}^\top \mathbf{A})^{-1} \mathbf{a}_i$, see Appendix A.4. This can be viewed as affecting the component-wise smoothness of the objective, which hinders subsampling-based estimators of the Hessian and the gradient.

In Figure 4(b), we illustrate how the performance of SVRN-HA and SN-HA degrades for the high-coherence least squares task, and we also show how this can be addressed by relying on the ideas developed in Section 2.2 (and further discussed in Appendix A.4). First, notice that not only is the convergence rate of both SVRN-HA and SN-HA worse on the high-coherence dataset than on the previous least squares examples (e.g., compare with Figure 3(b)), but also, the acceleration coming from variance reduction is drastically reduced to the point of being negligible. The former effect is primarily caused by the fact that uniform Hessian

subsampling is much less effective at producing accurate approximations for high-coherence matrices, and this affects both algorithms similarly (we note that one could construct an even more highly coherent matrix, for which these methods would essentially stop converging altogether). The latter effect is the consequence of the fact that gradient subsampling is also adversely affected by high coherence, so it becomes nearly impossible to produce gradient estimates with uniform sampling that would lead to an accelerated rate, even with variance reduction. This corresponds to the regime of $\kappa \geq n$ in our theory.

Fortunately, for least squares regression, this phenomenon can be addressed easily. As outlined in Appendix A.4, we can use one of two strategies: (1) use importance sampling proportional to the leverage scores of **A** for both the Hessian and gradient estimates; or (2) precondition the problem using the Randomized Hadamard Transform (RHT) to uniformize all the leverage scores, and then use uniform subsampling. Both of these methods require roughly $O(nd \log n)$ preprocessing cost and eliminate dependence on the condition number for both SVRN-HA and SN-HA. The latter strategy is somewhat more straightforward since it does not require modifying the optimization algorithms, and we apply it here for our high coherence least squares task: we let SVRN-HA-RHT and SN-HA-RHT denote the two optimization algorithms ran after applying the RHT preconditioning to the problem. Note that this not only improves the convergence rate of both methods but also brings back the accelerated rate enjoyed by SVRN-HA in the previous experiments. In fact, our least squares results (Theorem 2 and Lemma 6) can be directly applied to SVRN-HA-RHT, so this method and its accelerated convergence rate of $\tilde{O}(d/n)$ is provably unaffected by any high-coherence matrices.

## 6 Conclusions

We propose and analyze Stochastic Variance-Reduced Newton (SVRN), a provably effective strategy of incorporating variance reduction into popular stochastic Newton methods for solving finite-sum minimization tasks. We show that SVRN improves the local convergence complexity of Subsampled Newton (per data pass) from $O(\alpha \log(1/\epsilon))$ to $O\left(\frac{\log(1/\epsilon)}{\log(n)}\right)$, while retaining all the benefits of second-order optimization, such as a simple unit step size and easily scalable large-batch operations.

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

# A   Proofs for local convergence analysis of SVRN

In this section, we provide the proofs of our main technical results, i.e., local convergence analysis for SVRN. First, we prove the result for the general case (Theorem 3), then we prove the result for least squares (Theorem 2).

## A.1   Preliminaries

First, let us recall the formal definitions of the standard Hessian regularity assumptions used in Theorem 3. For all our results, it is sufficient that the function $f$ satisfies either one of these assumptions.

**Assumption 2** *Function $f : \mathbb{R}^d \to \mathbb{R}$ has Lipschitz continuous Hessian with constant $L$, i.e., $\|\nabla^2 f(\mathbf{x}) - \nabla^2 f(\mathbf{x}')\| \leq L \|\mathbf{x} - \mathbf{x}'\|$ for all $\mathbf{x}, \mathbf{x}' \in \mathbb{R}^d$.*

**Assumption 3** *Function $f : \mathbb{R}^d \to \mathbb{R}$ is self-concordant, i.e., for all $\mathbf{x}, \mathbf{x}' \in \mathbb{R}^d$, the function $\phi(t) = f(\mathbf{x}+t\mathbf{x}')$ satisfies: $|\phi'''(t)| \leq 2(\phi''(t))^{3/2}$.*

In the proof, we use the following version of Bernstein's concentration inequality for random vectors (Corollary 4.1 in Minsker, 2017).

**Lemma 1** *Let $\mathbf{v}_1, ..., \mathbf{v}_m \in \mathbb{R}^d$ be independent random vectors such that $\mathbb{E}[\mathbf{v}_i] = \mathbf{0}$ and $\|\mathbf{v}_i\| \leq R$ almost surely. Denote $\sigma^2 := \sum_{i=1}^m \mathbb{E} \|\mathbf{v}_i\|^2$. Then, for all $t^2 \geq \sigma^2 + tR/3$, we have*

$$\Pr\left\{ \Big\| \sum_{i=1}^m \mathbf{v}_i \Big\| > t \right\} \leq 28 \exp\Big( -\frac{t^2/2}{\sigma^2 + tR/3} \Big).$$

We also use the following lemma to convert from convergence in the norm, $\|\mathbf{x} - \mathbf{x}^*\|_{\mathbf{H}}$, to convergence in excess loss, $f(\mathbf{x}) - f(\mathbf{x}^*)$, in the neighborhood around the optimum $\mathbf{x}^*$. The proof of this lemma, given in Appendix A.3, uses Quadratic Taylor's Theorem.

**Lemma 2** *If $f$ satisfies Assumption 1 and either Assumption 2 or 3, then for any $\epsilon \in [0,1]$ and $\mathbf{x} \in U_f(\epsilon_l)$, we have:*

$$\nabla^2 f(\mathbf{x}) \approx_{\epsilon_l} \nabla^2 f(\mathbf{x}^*) \qquad and \qquad f(\mathbf{x}) - f(\mathbf{x}^*) \approx_{\epsilon_l} \frac{1}{2}\|\mathbf{x} - \mathbf{x}^*\|_{\nabla^2 f(\mathbf{x}^*)}^2.$$

## A.2 Proof of Theorem 3

To simplify the notation, we will drop the subscript $s$, so that $\tilde{\mathbf{x}} = \tilde{\mathbf{x}}_s$ and $\tilde{\mathbf{g}} = \tilde{\mathbf{g}}_s$. Also, let us define $\widehat{\mathbf{g}}(\mathbf{x}) = \frac{1}{m} \sum_{i=1}^{m} \nabla \psi_i(\mathbf{x})$. We use $\mathbf{g}(\mathbf{x}) = \nabla f(\mathbf{x})$, $\mathbf{g}_t = \mathbf{g}(\mathbf{x}_t)$, $\mathbf{H}_t = \nabla^2 f(\mathbf{x}_t)$, $\mathbf{H} = \nabla^2 f(\mathbf{x}^*)$, $\widehat{\mathbf{g}}_t = \widehat{\mathbf{g}}(\mathbf{x}_t)$, and $\bar{\mathbf{g}}_t = \widehat{\mathbf{g}}_t - \widehat{\mathbf{g}}(\tilde{\mathbf{x}}) + \tilde{\mathbf{g}}$ as shorthands. Also, we will use $\Delta_t = \mathbf{x}_t - \mathbf{x}^*$. We start by splitting up the error bound into two terms: the first one is an error term that would arise if we were using the exact gradient $\mathbf{g}_t$ instead of the gradient estimate $\bar{\mathbf{g}}_t$; and the second term addresses the error coming from the noise in the gradient estimate. Initially, we use the error $\|\Delta_t\|_{\mathbf{H}}$ to analyze the convergence rate in one step of the procedure, where recall that $\|\mathbf{v}\|_{\mathbf{M}} = \sqrt{\mathbf{v}^\top \mathbf{M} \mathbf{v}}$. We then convert that to get convergence in function value via Lemma 2.

We first address the assumption that $\tilde{\mathbf{H}}$ is an $\alpha$-approximation of $\mathbf{H}_t$, as defined by the condition (2). Note that, via Lemma 2, for any $\mathbf{x}_t \in U_f(\epsilon_l)$ we have that $\mathbf{H}_t \approx_{\epsilon_l} \mathbf{H}$, which for a sufficiently small $\epsilon_l$ implies that $\mathbf{H}_t$ is a 1.1-approximation of $\mathbf{H}$ in the sense of (2). This, in turn implies that $\tilde{\mathbf{H}}$ is a $1.1\alpha$-approximation of $\mathbf{H}$, because $\tilde{\mathbf{H}} \preceq \sqrt{\alpha}\mathbf{H}_t \preceq \sqrt{1.1\alpha}\mathbf{H}$ (the other direction is analogous). For the sake of simplicity, we will replace $1.1\alpha$ with $\alpha$ and say that $\tilde{\mathbf{H}}$ is an $\alpha$-approximation of $\mathbf{H}$ (this can be easily accounted for by adjusting the constants at the end).

Now, suppose that after $t$ inner iterations, we get $\mathbf{x}_t \in U_f(\epsilon_l)$ satisfying $\|\Delta_t\|_{\mathbf{H}} \leq \|\Delta_0\|_{\mathbf{H}}$. Our decomposition of the error into two terms proceeds as follows, where we use $\tilde{\mathbf{p}}_t = \tilde{\mathbf{H}}^{-1}\bar{\mathbf{g}}_t$:

$$
\begin{aligned}
\|\Delta_{t+1}\|_{\mathbf{H}} &= \|(\mathbf{x}_t - \eta\tilde{\mathbf{p}}_t) - \mathbf{x}^*\|_{\mathbf{H}} \\
&= \|\Delta_t - \eta\tilde{\mathbf{H}}^{-1}\mathbf{g}_t + \eta\tilde{\mathbf{H}}^{-1}\mathbf{g}_t - \eta\tilde{\mathbf{H}}^{-1}\bar{\mathbf{g}}_t\|_{\mathbf{H}} \\
&\leq \|\Delta_t - \eta\tilde{\mathbf{H}}^{-1}\mathbf{g}_t\|_{\mathbf{H}} + \eta\|\tilde{\mathbf{H}}^{-1}(\mathbf{g}_t - \bar{\mathbf{g}}_t)\|_{\mathbf{H}}
\end{aligned}
\tag{6}
$$

To bound the second term in (6), we first observe that $\tilde{\mathbf{H}}^{-1} \preceq \sqrt{\alpha}\mathbf{H}^{-1}$, which in turn yields $\mathbf{H}^{1/2}\tilde{\mathbf{H}}^{-1}\mathbf{H}^{1/2} \preceq \sqrt{\alpha}\,\mathbf{I}$. Thus, we can write $\|\mathbf{H}^{1/2}\tilde{\mathbf{H}}^{-1}\mathbf{H}^{1/2}\| \leq \sqrt{\alpha}$ and we get:

$$
\begin{aligned}
\|\tilde{\mathbf{H}}^{-1}(\mathbf{g}_t - \bar{\mathbf{g}}_t)\|_{\mathbf{H}} &= \|\mathbf{H}^{1/2}\tilde{\mathbf{H}}^{-1}\mathbf{H}^{1/2}\mathbf{H}^{-1/2}(\mathbf{g}_t - \bar{\mathbf{g}}_t)\| \\
&\leq \|\mathbf{H}^{1/2}\tilde{\mathbf{H}}^{-1}\mathbf{H}^{1/2}\| \cdot \|\mathbf{H}^{-1/2}(\mathbf{g}_t - \bar{\mathbf{g}}_t)\| \\
&\leq \sqrt{\alpha} \cdot \|\mathbf{g}_t - \bar{\mathbf{g}}_t\|_{\mathbf{H}^{-1}}
\end{aligned}
$$

We now break $\|\mathbf{g}_t - \bar{\mathbf{g}}_t\|_{\mathbf{H}^{-1}}$ down into two parts, introducing $\widehat{\mathbf{g}}(\mathbf{x}^*)$ and separating $\widehat{\mathbf{g}}(\tilde{\mathbf{x}})$ from $\widehat{\mathbf{g}}_t$:

$$
\begin{aligned}
\|\mathbf{g}_t - \bar{\mathbf{g}}_t\|_{\mathbf{H}^{-1}} &= \|\mathbf{g}_t - (\widehat{\mathbf{g}}_t - \widehat{\mathbf{g}}(\tilde{\mathbf{x}}) + \tilde{\mathbf{g}})\|_{\mathbf{H}^{-1}} \\
&\leq \|\mathbf{g}_t - (\widehat{\mathbf{g}}_t - \widehat{\mathbf{g}}(\mathbf{x}^*))\|_{\mathbf{H}^{-1}} + \|\tilde{\mathbf{g}} - (\widehat{\mathbf{g}}(\tilde{\mathbf{x}}) - \widehat{\mathbf{g}}(\mathbf{x}^*))\|_{\mathbf{H}^{-1}}.
\end{aligned}
$$

We bound the above two terms using the following lemma, which gives a new high-probability error bound for the variance reduced gradient estimates in the large mini-batch regime which, unlike results from prior work that hold in expectation, crucially relies on the iterate being in the local neighborhood $U_f(1)$ around the optimum $\mathbf{x}^*$.

**Lemma 3** *There is an absolute constant $C > 0$ such that for any $\mathbf{x} \in U_f(1)$, letting $\mathbf{H} = \nabla^2 f(\mathbf{x}^*)$, the gradient estimate $\widehat{\mathbf{g}}(\mathbf{x}) = \frac{1}{m}\sum_{i=1}^{m} \nabla\psi_i(\mathbf{x})$ using $m \geq \kappa \log(1/\delta)$ samples, with probability $1 - \delta$ satisfies:*

$$
\left\|\widehat{\mathbf{g}}(\mathbf{x}) - \widehat{\mathbf{g}}(\mathbf{x}^*) - \nabla f(\mathbf{x})\right\|_{\mathbf{H}^{-1}}^2 \leq C \log(1/\delta)\frac{\kappa}{m}\|\mathbf{x} - \mathbf{x}^*\|_{\mathbf{H}}^2.
$$

**Proof** We will apply Bernstein's concentration inequality for random vectors (Lemma 1) to $\mathbf{v}_i = \nabla\psi_i(\mathbf{x}) - \nabla\psi_i(\mathbf{x}^*) - \nabla f(\mathbf{x})$. First, observe that $\mathbb{E}\,\nabla\psi_i(\mathbf{x}) = \nabla f(\mathbf{x})$ and $\mathbb{E}\,\nabla\psi_i(\mathbf{x}^*) = \nabla f(\mathbf{x}^*) = \mathbf{0}$, so in particular, $\mathbb{E}[\mathbf{v}_i] = \mathbf{0}$.

In the next step, we will use the fact that for any $\lambda$-smooth function $g$, we have $\|\nabla g(\mathbf{x})\|^2 \leq 2\lambda \cdot (g(\mathbf{x}) - \min_{\mathbf{x}'} g(\mathbf{x}'))$, which follows because:

$$
\begin{aligned}
\min_{\mathbf{x}'} g(\mathbf{x}') &\leq g\big(\mathbf{x} - \tfrac{1}{\lambda}\nabla g(\mathbf{x})\big) \\
&\leq g(\mathbf{x}) - \frac{1}{\lambda}\|\nabla g(\mathbf{x})\|^2 + \frac{\lambda}{2}\frac{1}{\lambda^2}\|\nabla g(\mathbf{x})\|^2 \\
&= g(\mathbf{x}) - \frac{1}{2\lambda}\|\nabla g(\mathbf{x})\|^2.
\end{aligned}
$$

We will use this fact once on $f$, and also a second time, on the function $g(\mathbf{x}) = \psi_i(\mathbf{x}) - \psi_i(\mathbf{x}^*) - (\mathbf{x} - \mathbf{x}^*)^\top \nabla \psi(\mathbf{x}^*)$, which is $\lambda$-smooth because $\psi_i$ is $\lambda$-smooth, observing that $\nabla g(\mathbf{x}) = \nabla \psi_i(\mathbf{x}) - \nabla \psi_i(\mathbf{x}^*)$ and that $\min_{\mathbf{x}'} g(\mathbf{x}') = g(\mathbf{x}^*) = 0$. Thus, we have

$$\begin{aligned}
\|\mathbf{v}_i\|^2 &\leq 2\|\nabla \psi_i(\mathbf{x}) - \nabla \psi_i(\mathbf{x}^*)\|^2 + 2\|\nabla f(\mathbf{x})\|^2 \\
&\leq 4\lambda \cdot \big(\psi_i(\mathbf{x}) - \psi_i(\mathbf{x}^*) - (\mathbf{x} - \mathbf{x}^*)^\top \nabla \psi(\mathbf{x}^*)\big) + 4\lambda \cdot \big(f(\mathbf{x}) - f(\mathbf{x}^*)\big) \\
&\leq 2\lambda^2 \|\mathbf{x} - \mathbf{x}^*\|^2 + 2\lambda^2 \|\mathbf{x} - \mathbf{x}^*\|^2 = 4\lambda^2 \|\mathbf{x} - \mathbf{x}^*\|^2,
\end{aligned}$$

where in the last step we used again that $\psi_i$ and $f$ are $\lambda$-smooth. To bound the expectation $\mathbb{E}\|\mathbf{v}_i\|^2$, we use the intermediate inequality from the above derivation, obtaining:

$$\begin{aligned}
\mathbb{E}\|\mathbf{v}_i\|^2 &= \mathbb{E}\big[\|\nabla \psi_i(\mathbf{x}) - \nabla \psi_i(\mathbf{x}^*)\|^2\big] - 2\,\mathbb{E}\big[\nabla \psi_i(\mathbf{x}) - \nabla \psi_i(\mathbf{x}^*)\big]^\top \nabla f(\mathbf{x}) + \|\nabla f(\mathbf{x})\|^2 \\
&= \mathbb{E}\big[\|\nabla \psi_i(\mathbf{x}) - \nabla \psi_i(\mathbf{x}^*)\|^2\big] - \|\nabla f(\mathbf{x})\|^2 \\
&\leq \mathbb{E}\big[\|\nabla \psi_i(\mathbf{x}) - \nabla \psi_i(\mathbf{x}^*)\|^2\big] \\
&\leq \mathbb{E}\Big[2\lambda \cdot \big(\psi_i(\mathbf{x}) - \psi_i(\mathbf{x}^*) - (\mathbf{x} - \mathbf{x}^*)^\top \nabla \psi_i(\mathbf{x}^*)\big)\Big] \\
&= 2\lambda \cdot \big(f(\mathbf{x}) - f(\mathbf{x}^*) - (\mathbf{x} - \mathbf{x}^*)\nabla f(\mathbf{x}^*)\big) \\
&= 2\lambda \cdot \big(f(\mathbf{x}) - f(\mathbf{x}^*)\big).
\end{aligned}$$

We now use the assumption that $\mathbf{x} \in U_f(1)$, which implies via Lemma 2 that $f(\mathbf{x}) - f(\mathbf{x}^*) \leq 2 \cdot \frac{1}{2}\|\mathbf{x} - \mathbf{x}^*\|_{\mathbf{H}}^2 = \|\mathbf{x} - \mathbf{x}^*\|_{\mathbf{H}}^2$. Thus, we can use Lemma 1 with $R = 2\lambda\|\mathbf{x} - \mathbf{x}^*\|$ and $\sigma^2 = 2m\lambda\|\mathbf{x} - \mathbf{x}^*\|_{\mathbf{H}}^2$, as well as $\mu$-strong convexity of $f$, obtaining that, for some absolute constant $C$, with probability $1 - \delta$, we have:

$$\begin{aligned}
\|\widehat{\mathbf{g}}(\mathbf{x}) - \widehat{\mathbf{g}}(\mathbf{x}^*) - \nabla f(\mathbf{x})\|_{\mathbf{H}^{-1}}^2 &\leq \frac{1}{\mu} \Big\| \frac{1}{m} \sum_{i=1}^{m} \mathbf{v}_i \Big\|^2 \\
&\leq \frac{C}{\mu} \Big( \frac{\sigma^2 \log(1/\delta)}{m^2} + \frac{R^2 \log^2(1/\delta)}{m^2} \Big) \\
&\leq \frac{C}{\mu} \Big( \frac{2\lambda\|\mathbf{x} - \mathbf{x}^*\|_{\mathbf{H}}^2 \log(1/\delta)}{m} + \frac{4\lambda^2\|\mathbf{x} - \mathbf{x}^*\|^2 \log^2(1/\delta)}{m^2} \Big) \\
&\leq 4C \Big( \frac{\kappa \log(1/\delta)}{m} + \frac{\kappa^2 \log^2(1/\delta)}{m^2} \Big) \cdot \|\mathbf{x} - \mathbf{x}^*\|_{\mathbf{H}}^2 \\
&\leq 8C \log(1/\delta) \cdot \frac{\kappa}{m} \|\mathbf{x} - \mathbf{x}^*\|_{\mathbf{H}}^2,
\end{aligned}$$

where in the last step we used that $m \geq \kappa \log(1/\delta)$. ∎

Letting $\epsilon_g = \sqrt{2C \log(t/\delta)\kappa/m}$, Lemma 3 implies that with probability $1 - \delta/t^2$,

$$\|\mathbf{g}_t - (\widehat{\mathbf{g}}_t - \widehat{\mathbf{g}}(\tilde{\mathbf{x}}) + \tilde{\mathbf{g}})\|_{\mathbf{H}^{-1}} \leq \epsilon_g \big(\|\Delta_t\|_{\mathbf{H}} + \|\Delta_0\|_{\mathbf{H}}\big) \leq 2\epsilon_g \|\Delta_0\|_{\mathbf{H}}.$$

Finally, we return to the first term in (6), i.e., $\|\Delta_t - \eta \tilde{\mathbf{H}}^{-1} \mathbf{g}_t\|_{\mathbf{H}}$. To control this term we introduce the following lemma which is potentially of independent interest to the local convergence analysis of Newton-type methods.

**Lemma 4** *Suppose that $f$ satisfies Assumption 1 and either one of the Assumptions 2 or 3, and take any $\mathbf{x} \in U_f(\epsilon_l)$ (see Definition 1) for $\epsilon_l \leq 1/c\alpha$ for a sufficiently large absolute constant $c > 0$. Let $\mathbf{H} = \nabla^2 f(\mathbf{x}^*)$ and consider a pd matrix $\tilde{\mathbf{H}}$ that satisfies $\frac{1}{\sqrt{\alpha}} \mathbf{H} \preceq \tilde{\mathbf{H}} \preceq \sqrt{\alpha}\,\mathbf{H}$. Then, for $\eta := \min\{\sqrt{2/\alpha}, 1\}$, we have:*

$$\|\mathbf{x} - \eta \tilde{\mathbf{H}}^{-1} \nabla f(\mathbf{x}) - \mathbf{x}^*\|_{\mathbf{H}} \leq \Big(1 - \frac{1}{1.9\alpha}\Big) \|\mathbf{x} - \mathbf{x}^*\|_{\mathbf{H}}.$$

**Proof** Let $\Delta_0 := \mathbf{x} - \mathbf{x}^*$ and $\Delta_1 := \mathbf{x} - \eta\tilde{\mathbf{H}}^{-1}\nabla f(\mathbf{x}) - \mathbf{x}^*$. Using that $\nabla f(\mathbf{x}^*) = \mathbf{0}$, we have:

$$
\begin{aligned}
\Delta_1 &= \Delta_0 - \eta\tilde{\mathbf{H}}^{-1}\nabla f(\mathbf{x}) \\
&= \Delta_0 - \eta\tilde{\mathbf{H}}^{-1}(\nabla f(\mathbf{x}) - \nabla f(\mathbf{x}^*)) \\
&= \Delta_0 - \eta\tilde{\mathbf{H}}^{-1}\int_0^1 \nabla^2 f(\mathbf{x}^* + \theta\Delta_0)\Delta_0 d\theta \\
&= (\mathbf{I} - \eta\tilde{\mathbf{H}}^{-1}\bar{\mathbf{H}})\Delta_0,
\end{aligned}
$$

where we defined $\bar{\mathbf{H}} := \int_0^1 \nabla^2 f(\mathbf{x}^* + \theta\Delta_0)d\theta$. It follows that we can bound the norm of $\Delta_1$ using a norm defined by the matrix $\bar{\mathbf{H}}$:

$$
\begin{aligned}
\|\Delta_1\|_{\bar{\mathbf{H}}} &= \|\bar{\mathbf{H}}^{1/2}(\mathbf{I} - \eta\tilde{\mathbf{H}}^{-1}\bar{\mathbf{H}})\Delta_0\| \\
&= \|(\mathbf{I} - \eta\mathbf{H}^{1/2}\tilde{\mathbf{H}}^{-1}\bar{\mathbf{H}}^{1/2})\bar{\mathbf{H}}^{1/2}\Delta_0\| \\
&\leq \|\mathbf{I} - \eta\bar{\mathbf{H}}^{1/2}\tilde{\mathbf{H}}^{-1}\bar{\mathbf{H}}^{1/2}\| \cdot \|\Delta_0\|_{\bar{\mathbf{H}}}.
\end{aligned}
$$

Observe that for any $\theta \in [0,1]$, the vector $\mathbf{x}^* + \theta\Delta_0$ belongs to $U_f(\epsilon_l)$, which via Lemma 2 implies that

$$
\nabla^2 f(\mathbf{x}^* + \theta\Delta_0) \approx_{\epsilon_l} \mathbf{H} \quad \forall \theta \in [0,1].
$$

where $\mathbf{H} = \nabla^2 f(\mathbf{x}^*)$. In particular, this means that $\bar{\mathbf{H}} \approx_{\epsilon_l} \mathbf{H}$, which, combined with the $\alpha$-approximation property of $\tilde{\mathbf{H}}$, allows us to write the following:

$$
\tilde{\mathbf{H}}^{-1} \preceq \sqrt{\alpha}\mathbf{H}^{-1} \preceq \sqrt{\alpha}(1 + \epsilon_l)\bar{\mathbf{H}}^{-1} \quad \text{and} \quad \tilde{\mathbf{H}}^{-1} \succeq \frac{1}{\sqrt{\alpha}}\mathbf{H}^{-1} \succeq \frac{1 - \epsilon_l}{\sqrt{\alpha}}\bar{\mathbf{H}}^{-1}.
$$

Putting these inequalities together, we obtain that:

$$
\eta\frac{1 - \epsilon_l}{\sqrt{\alpha}}\mathbf{I} \preceq \eta\bar{\mathbf{H}}^{1/2}\tilde{\mathbf{H}}^{-1}\bar{\mathbf{H}}^{1/2} \preceq \eta\sqrt{\alpha}(1 + \epsilon_l)\mathbf{I}.
$$

Now, using the fact that $\eta = \min\{\sqrt{2/\alpha}, 1\}$, we conclude that:

$$
\begin{aligned}
\|\mathbf{I} - \eta\bar{\mathbf{H}}^{1/2}\tilde{\mathbf{H}}^{-1}\bar{\mathbf{H}}^{1/2}\| &\leq \max\left\{\eta\sqrt{\alpha}(1 + \epsilon_l) - 1, 1 - \eta\frac{1 - \epsilon_l}{\sqrt{\alpha}}\right\} \\
&\leq \max\left\{\sqrt{2}(1 + \epsilon_l) - 1, 1 - \frac{\sqrt{2}(1 - \epsilon_l)}{\alpha}, 1 - \frac{1 - \epsilon_l}{\sqrt{2}}\right\} \\
&\leq \max\left\{1 - \frac{1}{1.8}, 1 - \frac{1}{\alpha}\right\} \leq 1 - \frac{1}{1.8\alpha},
\end{aligned}
$$

where we used that $\epsilon_l \leq 1/c$ for a sufficiently large constant $c > 0$ such that $\max\{\sqrt{2}(1 + \epsilon_l) - 1, 1 - \frac{1 - \epsilon_l}{\sqrt{2}}\} \leq 1 - \frac{1}{1.8}$. Now, we analyze convergence in the norm induced by $\mathbf{H}$, instead of $\bar{\mathbf{H}}$, by relying again on the fact that $\bar{\mathbf{H}} \approx_{\epsilon_l} \mathbf{H}$, obtaining:

$$
\begin{aligned}
\|\Delta_1\|_{\mathbf{H}} &\leq \frac{1}{\sqrt{1 - \epsilon_l}}\|\Delta_1\|_{\bar{\mathbf{H}}} \leq \frac{1}{\sqrt{1 - \epsilon_l}}\left(1 - \frac{1}{1.8\alpha}\right)\|\Delta_0\|_{\bar{\mathbf{H}}} \\
&\leq \sqrt{\frac{1 + \epsilon_l}{1 - \epsilon_l}}\left(1 - \frac{1}{1.8\alpha}\right)\|\Delta_0\|_{\mathbf{H}} \leq \left(1 - \frac{1}{1.9\alpha}\right)\|\Delta_0\|_{\mathbf{H}},
\end{aligned}
$$

where the last step requires $\epsilon_l = 1/c\alpha$ for sufficiently large absolute constant $c > 0$. ∎

Using Lemma 4 to bound the first term in (6), we obtain that:

$$
\|\Delta_t - \eta\tilde{\mathbf{H}}_t^{-1}\mathbf{g}_t\|_{\mathbf{H}} \leq \left(1 - \frac{1}{1.9\alpha}\right)\|\Delta_t\|_{\mathbf{H}}.
$$

Putting everything together, we obtain the following bound for the error of the update that uses the stochastic variance-reduced gradient estimate:

$$
\begin{aligned}
\|\Delta_{t+1}\|_{\mathbf{H}} &= \|\Delta_t - \eta\tilde{\mathbf{p}}_t\|_{\mathbf{H}} \\
&\leq \|\Delta_t - \eta\tilde{\mathbf{H}}^{-1}\mathbf{g}_t\|_{\mathbf{H}} + \eta\|\tilde{\mathbf{H}}^{-1}(\mathbf{g}_t - \bar{\mathbf{g}}_t)\|_{\mathbf{H}} \\
&\leq \left(1 - \frac{1}{1.9\alpha}\right)\|\Delta_t\|_{\mathbf{H}} + 2\eta\sqrt{\alpha}\epsilon_g\|\Delta_0\|_{\mathbf{H}} \\
&\leq \left(1 - \frac{1}{1.9\alpha}\right)\|\Delta_t\|_{\mathbf{H}} + 3\epsilon_g\|\Delta_0\|_{\mathbf{H}}.
\end{aligned}
$$

Note that, as long as $3\epsilon_g \leq \frac{1}{2\alpha}$ (which can be ensured by our assumption on $m$), this implies that $\|\Delta_{t+1}\|_{\mathbf{H}} \leq \|\Delta_0\|_{\mathbf{H}}$ and so $\mathbf{x}_{t+1} \in U_f(\epsilon_l)$. Thus, our analysis can be applied recursively at each inner iteration. To expand the error recursion, observe that if we apply a union bound over the high-probability events in Lemma 3 for each inner iteration $t$ using failure probability $\delta_t = \delta/t^2$, then they hold for all $t$ with probability at least $1 - \sum_{t=1}^{\infty} \delta/t^2 \geq 1 - \delta\pi^2/6$. We obtain:

$$
\begin{aligned}
\|\Delta_t\|_{\mathbf{H}} &\leq \left(1 - \frac{1}{1.9\alpha}\right)\|\Delta_{t-1}\|_{\mathbf{H}} + 3\epsilon_g\|\Delta_0\|_{\mathbf{H}} \\
&\leq \left(1 - \frac{1}{1.9\alpha}\right)^t\|\Delta_0\|_{\mathbf{H}} + \left(\sum_{i=0}^{t-1}\left(1 - \frac{1}{1.9\alpha}\right)^i\right)\cdot 3\epsilon_g\|\Delta_0\|_{\mathbf{H}} \\
&\leq \left(\left(1 - \frac{1}{1.9\alpha}\right)^t + 9\alpha\epsilon_g\right)\cdot\|\Delta_0\|_{\mathbf{H}}.
\end{aligned}
$$

Applying Lemma 2, we convert this to convergence in function value:

$$
\begin{aligned}
f(\mathbf{x}_t) - f(\mathbf{x}^*) &\leq \frac{1}{1-\epsilon_l}\frac{1}{2}\|\Delta_t\|_{\mathbf{H}}^2 \\
&\leq \frac{1}{1-\epsilon_l}\frac{1}{2}\left(\left(1 - \frac{1}{1.9\alpha}\right)^t + 9\alpha\epsilon_g\right)^2\|\Delta_0\|_{\mathbf{H}}^2 \\
&\leq \frac{1+\epsilon_l}{1-\epsilon_l}\left(\left(1 - \frac{1}{1.9\alpha}\right)^{2t} + 9^2\alpha^2\epsilon_g^2\right)\cdot(f(\mathbf{x}_0) - f(\mathbf{x}^*)) \\
&\leq \left(\left(1 - \frac{1}{2\alpha}\right)^t + C'\alpha^2\frac{\kappa\log(t/\delta)}{m}\right)\cdot(f(\mathbf{x}_0) - f(\mathbf{x}^*)),
\end{aligned}
$$

where $C'$ is an absolute constant, and we again used that $\epsilon_l \leq 1/c\alpha$ for a sufficiently large $c$, thus concluding the proof.

### A.3   Proof of Lemma 2

First, we show that the Hessian at $\mathbf{x} \in U_f(\epsilon_l)$ is an $\epsilon_l$-approximation of the Hessian at the optimum $\mathbf{x}^*$. This is broken down into two cases, depending on which of the two Assumptions 2 and 3 are satisfied.

Case 1: Assumption 2 (Lipschitz Hessian).   Using the shorthand $\mathbf{H} = \nabla^2 f(\mathbf{x}^*)$ and the fact that strong convexity (Assumption 1) implies that $\nabla^2 f(\mathbf{x}) \succeq \mu\mathbf{I}$, we have:

$$
\|\mathbf{H}^{-1/2}(\nabla^2 f(\mathbf{x}) - \mathbf{H})\mathbf{H}^{-1/2}\| \leq \frac{1}{\mu}\|\nabla^2 f(\mathbf{x}) - \mathbf{H}\| \leq \frac{L}{\mu}\|\mathbf{x} - \mathbf{x}^*\| \leq \frac{L}{\mu^{3/2}}\|\mathbf{x} - \mathbf{x}^*\|_{\mathbf{H}} \leq \epsilon_l,
$$

which implies that $\nabla^2 f(\mathbf{x}) \approx_{\epsilon_l} \mathbf{H}$.

Case 2: Assumption 3 (Self-concordance).   The fact that $\nabla^2 f(\mathbf{x}) \approx_{\epsilon_l} \mathbf{H}$ follows from the following property of self-concordant functions (Boyd & Vandenberghe, 2004, Chapter 9.5), which holds when $\|\mathbf{x} - \mathbf{x}^*\|_{\mathbf{H}} < 1$:

$$
(1 - \|\mathbf{x} - \mathbf{x}^*\|_{\mathbf{H}})^2 \cdot \mathbf{H} \preceq \nabla^2 f(\mathbf{x}) \preceq (1 - \|\mathbf{x} - \mathbf{x}^*\|_{\mathbf{H}})^{-2} \cdot \mathbf{H},
$$

where we again let $\mathbf{H} = \nabla^2 f(\mathbf{x}^*)$.

We next use a version of Quadratic Taylor's Theorem, as given below. See Theorem 3 in Chapter 2.6 of (Jerrard, 2018) and Chapter 2.7 in (Folland, 2002).

**Lemma 5** *Suppose that $f : \mathbb{R}^d \to \mathbb{R}$ has continuous first and second derivatives. Then, for any $\mathbf{a}$ and $\mathbf{v}$, there exists $\theta \in (0, 1)$ such that:*

$$f(\mathbf{a} + \mathbf{v}) = f(\mathbf{a}) + \nabla f(\mathbf{a})^\top \mathbf{v} + \frac{1}{2}\mathbf{v}^\top \nabla^2 f(\mathbf{a} + \theta\mathbf{v})\mathbf{v}.$$

Applying Talyor's theorem with $\mathbf{a} = \mathbf{x}^*$ and $\mathbf{v} = \mathbf{x} - \mathbf{x}^*$, there is a $\mathbf{z} = \mathbf{x}^* + \theta(\mathbf{x} - \mathbf{x}^*)$ such that:

$$f(\mathbf{x}) - f(\mathbf{x}^*) = \frac{1}{2}\|\mathbf{x} - \mathbf{x}^*\|^2_{\nabla^2 f(\mathbf{z})},$$

where we use that $\nabla f(\mathbf{x}^*) = \mathbf{0}$. Since we assumed that $\mathbf{x} \in U_f(\epsilon_l)$, and naturally also $\mathbf{x}^* \in U_f(\epsilon_l)$, this means that $\mathbf{z} \in U_f(\epsilon_l)$, given that $U_f(\epsilon_l)$ is convex. Thus, using that $\nabla^2 f(\mathbf{z}) \approx_{\epsilon_l} \mathbf{H}$, we have $\|\mathbf{x} - \mathbf{x}^*\|^2_{\nabla^2 f(\mathbf{z})} \approx_{\epsilon_l} \|\mathbf{x} - \mathbf{x}^*\|^2_{\mathbf{H}}$.

### A.4 Proof of Theorem 2

In this section, we discuss how the convergence analysis of SVRN can be adapted to using leverage score sampling when solving a least squares task (proving Theorem 2).

Consider an expected risk minimization problem $f(\mathbf{x}) = \mathbb{E}[\psi(\mathbf{x})]$, where $\psi = \frac{1}{np_I}\psi_I$ and $I$ is an index from $\{1, ..., n\}$, sampled according to some importance sampling distribution $p$. More specifically, consider a least squares task, where the components are given by $\psi_i(\mathbf{x}) = \frac{1}{2}(\mathbf{a}_i^\top \mathbf{x} - y_i)^2$. Then, the overall minimization task becomes:

$$\mathbb{E}[\psi(\mathbf{x})] = \mathbb{E}\left[\frac{1}{np_I}\psi_I(\mathbf{x})\right] = \frac{1}{2n}\sum_{i=1}^n (\mathbf{a}_i^\top \mathbf{x} - y_i)^2. \tag{7}$$

Moreover, we have $f(\mathbf{x}) - f(\mathbf{x}^*) = \frac{1}{2n}\|\mathbf{A}(\mathbf{x} - \mathbf{x}^*)\|^2 = \frac{1}{2}\|\mathbf{x} - \mathbf{x}^*\|^2_{\mathbf{H}}$, where $\mathbf{H} = \nabla^2 f(\mathbf{x}) = \frac{1}{n}\mathbf{A}^\top \mathbf{A}$. Also,

$$\nabla\psi_i(\mathbf{x}) = (\mathbf{a}_i^\top \mathbf{x} - y_i)\mathbf{a}_i, \qquad \nabla^2\psi_i(\mathbf{x}) = \mathbf{a}_i \mathbf{a}_i^\top.$$

Naturally, since the Hessian is the same everywhere for this task, the local convergence neighborhood $U_f$ is simply the entire Euclidean space $\mathbb{R}^d$. Let us first recall our definition of the condition number for this task. Assumption 1 states that each $\psi_i$ is $\lambda$-smooth, i.e., $\|\nabla^2\psi_i(\mathbf{x})\| = \|\mathbf{a}_i\|^2 \leq \lambda$ and $f$ is $\mu$-strongly convex, i.e., $\lambda_{\min}(\mathbf{H}) = \frac{1}{n}\sigma^2_{\min}(\mathbf{A}) \geq \mu$, and the condition number of the problem is defined as $\kappa = \lambda/\mu \geq \max_i\{n\|\mathbf{a}_i\|^2\}/\sigma^2_{\min}(\mathbf{A})$. Can we reduce the condition number of this problem by importance sampling?

Consider the following naive strategy which can be applied directly with our convergence result. Here, we let the importance sampling probabilities be $p_i \propto \|\mathbf{a}_i\|^2$, so that the smoothness of the new reweighted problem will be $\tilde{\lambda} = \frac{1}{n}\sum_{i=1}^n \|\mathbf{a}_i\|^2$. In other words, it will be the average smoothness of the original problem, instead of the worst-case smoothness. Such importance sampling strategy can theoretically be applied to a general finite-sum minimization task with some potential gain, however we may need different sampling probabilities at each step. For least squares, the resulting condition number is $\tilde{\kappa} = \tilde{\lambda}/\mu = (\sum_i \|\mathbf{a}_i\|^2)/\sigma^2_{\min}(\mathbf{A})$. This is still worse than what we claimed for least squares, but it is still potentially much better than $\kappa$.

Next, we will show that by slightly adapting our convergence analysis, we can use leverage score sampling to further improve the convergence of SVRN for the least squares task. Recall that the $i$th leverage score of $\mathbf{A}$ is defined as $\ell_i = \|\mathbf{a}_i\|^2_{(\mathbf{A}^\top \mathbf{A})^{-1}} = \frac{1}{n}\|\mathbf{a}_i\|^2_{\mathbf{H}^{-1}}$, and the laverage scores satisfy $\sum_{i=1}^n \ell_i = d$. This result will require showing a specialized version of Lemma 3, which bounds the error in the variance-reduced subsampled gradient. In this case we show a bound in expectation, instead of with high probability.

**Lemma 6** *Suppose that $f$ defines a least squares task (7) and the sampling probabilities satisfy $p_i \geq \|\mathbf{a}_i\|^2_{(\mathbf{A}^\top \mathbf{A})^{-1}}/(Cd)$. Then, $\widehat{\mathbf{g}}(\mathbf{x}) = \frac{1}{m}\sum_{i=1}^m \frac{1}{np_{I_i}}\nabla\psi_{I_i}(\mathbf{x})$, where $I_1, ..., I_m \sim p$, satisfies:*

$$\mathbb{E}\|\widehat{\mathbf{g}}(\mathbf{x}) - \widehat{\mathbf{g}}(\mathbf{x}^*) - \nabla f(\mathbf{x})\|^2_{\mathbf{H}^{-1}} \leq C\frac{d}{m} \cdot \|\mathbf{x} - \mathbf{x}^*\|^2_{\mathbf{H}}.$$

**Proof** We define $\mathbf{v}_i = \frac{1}{np_{I_i}}(\nabla\psi_{I_i}(\mathbf{x}) - \nabla\psi_{I_i}(\mathbf{x}^*)) - \nabla f(\mathbf{x})$. Note that $\mathbb{E}[\mathbf{v}_i] = \mathbf{0}$, so we have:

$$\mathbb{E}\left\|\widehat{\mathbf{g}}(\mathbf{x}) - \widehat{\mathbf{g}}(\mathbf{x}^*) - \nabla f(\mathbf{x})\right\|_{\mathbf{H}^{-1}}^2 = \mathbb{E}\left\|\frac{1}{m}\sum_{i=1}^m \mathbf{v}_i\right\|_{\mathbf{H}^{-1}}^2 = \frac{1}{m}\mathbb{E}\left\|\mathbf{v}_1\right\|_{\mathbf{H}^{-1}}^2$$

$$\leq \frac{1}{m}\mathbb{E}\frac{1}{n^2 p_{I_1}^2}\|\nabla\psi_{I_1}(\mathbf{x}) - \nabla\psi_{I_1}(\mathbf{x}^*)\|_{\mathbf{H}^{-1}}^2$$

$$= \frac{1}{m}\mathbb{E}\frac{\|\mathbf{a}_{I_1}\|_{\mathbf{H}^{-1}}^2}{n^2 p_{I_1}^2}\left(\mathbf{a}_{I_1}^\top(\mathbf{x} - \mathbf{x}^*)\right)^2$$

$$\leq \frac{1}{m}Cd\cdot\mathbb{E}\frac{(\mathbf{a}_{I_1}^\top(\mathbf{x} - \mathbf{x}^*))^2}{np_{I_1}} = C\cdot\frac{d}{m}\|\mathbf{x} - \mathbf{x}^*\|_{\mathbf{H}}^2,$$

where we used that $\|\mathbf{a}_i\|_{\mathbf{H}^{-1}}^2 = n\|\mathbf{a}_i\|_{(\mathbf{A}^\top\mathbf{A})^{-1}}^2 \leq Cndp_i$. ∎

Since the above bound is obtained in expectation, to insert it into our high probability analysis, we apply Markov's inequality. Namely, it holds with probability $1 - \delta$ that:

$$\|\widehat{\mathbf{g}}(\mathbf{x}) - \widehat{\mathbf{g}}(\mathbf{x}^*) - \nabla f(\mathbf{x})\|_{\mathbf{H}^{-1}}^2 \leq \frac{Cd}{\delta m}\|\mathbf{x} - \mathbf{x}^*\|_{\mathbf{H}}^2.$$

Compared to Lemma 3, the dependence on the condition number $\kappa$ is completely eliminated in this result. Letting $m = n/\log(n/d)$ and the number of local iterations of SVRN to be $t = O(\log(n/d))$, we can apply the union bound argument from the proof of Theorem 3 by letting $\delta = 1/(Ct)$, so that with probability $1 - 1/C$, one stage of leverage score sampled SVRN satisfies:

$$f(\tilde{\mathbf{x}}_{s+1}) - f(\mathbf{x}^*) \leq \rho\cdot\left(f(\tilde{\mathbf{x}}_s) - f(\mathbf{x}^*)\right) \qquad \text{for} \qquad \rho = O\left(\frac{d\log^2(n/d)}{n}\right).$$

Alternatively, our main convergence analysis can be adapted (for least squares) to convergence in expectation, obtaining that $\mathbb{E}[f(\tilde{\mathbf{x}}_{s+1}) - f(\mathbf{x}^*)] \leq \tilde{\rho}\cdot\mathbb{E}[f(\tilde{\mathbf{x}}_s) - f(\mathbf{x}^*)]$ for $\tilde{\rho} = O(d\log(n/d)/n)$.

The time complexity stated in Theorem 2 comes from the fact that constructing a preconditioning matrix $\tilde{\mathbf{H}}$ that is an $\alpha$-approximation of $\mathbf{H}$ with $\alpha = O(1)$, together with approximating the leverage scores, takes $O(nd\log n + d^3\log d)$ (Drineas et al., 2012), whereas one stage of SVRN takes $O(nd + d^2\log(n/d))$. Here, the preconditioning matrix can be formed by applying a $k \times n$ sketching transformation $\mathbf{S}$ to the data matrix $\mathbf{A}$, and then computing the Hessian estimate $\frac{1}{n}\mathbf{A}^\top\mathbf{S}^\top\mathbf{S}\mathbf{A} \approx \mathbf{H}$. For example, if we use the Subsampled Randomized Hadamard Transform (SRHT, Ailon & Chazelle, 2009), then it suffices to use $k = O(d\log d)$. Finally, the initial iterate $\tilde{\mathbf{x}}_0$ can be constructed using the same sketching transformation via the so-called sketch-and-solve technique (Sarlos, 2006):

$$\tilde{\mathbf{x}}_0 = \operatorname*{argmin}_{\mathbf{x}}\|\mathbf{S}\mathbf{A}\mathbf{x} - \mathbf{S}\mathbf{y}\|^2.$$

With $k = O(d\log d)$, this initial iterate will satisfy $f(\tilde{\mathbf{x}}_0) \leq O(1)\cdot f(\mathbf{x}^*)$, so the number of iterations of SVRN needed to obtain $f(\tilde{\mathbf{x}}_s) \leq (1 + \epsilon)f(\mathbf{x}^*)$ is only $s = O\left(\frac{\log(1/\epsilon)}{\log(n/d)}\right)$.

We note that another way to implement SVRN with approximate leverage score sampling is to first precondition the entire least squares problem with a Randomized Hadamard Transform (i.e., SRHT without the subsampling):

$$\tilde{\mathbf{A}} = \mathbf{H}\mathbf{D}\mathbf{A} \qquad \text{and} \qquad \tilde{\mathbf{y}} = \mathbf{H}\mathbf{D}\mathbf{y}, \tag{8}$$

where $\mathbf{H}$ is a Hadamard matrix scaled by $1/\sqrt{n}$ and $\mathbf{D}$ is a diagonal matrix with random sign entries. This is a popular technique in Randomized Numerical Linear Algebra (Woodruff, 2014; Drineas & Mahoney, 2016; Dereziński & Mahoney, 2021). The cost of this transformation is $O(nd\log n)$, thanks to fast Fourier transform techniques, and the resulting least squares task is equivalent to the original one, because $\|\tilde{\mathbf{A}}\mathbf{x} - \tilde{\mathbf{y}}\|^2 = \|\mathbf{A}\mathbf{x} - \mathbf{y}\|^2$ for all $\mathbf{x}$. Moreover, with high probability, all of the leverage scores of $\tilde{\mathbf{A}}$ are nearly uniform, so, after this preconditioning, we can simply implement SVRN with uniform gradient subsampling and still enjoy the condition-number-free convergence rate from Theorem 2. This strategy is as efficient as direct leverage score sampling when $\mathbf{A}$ is a dense matrix, but it is less effective when we want to exploit data sparsity.

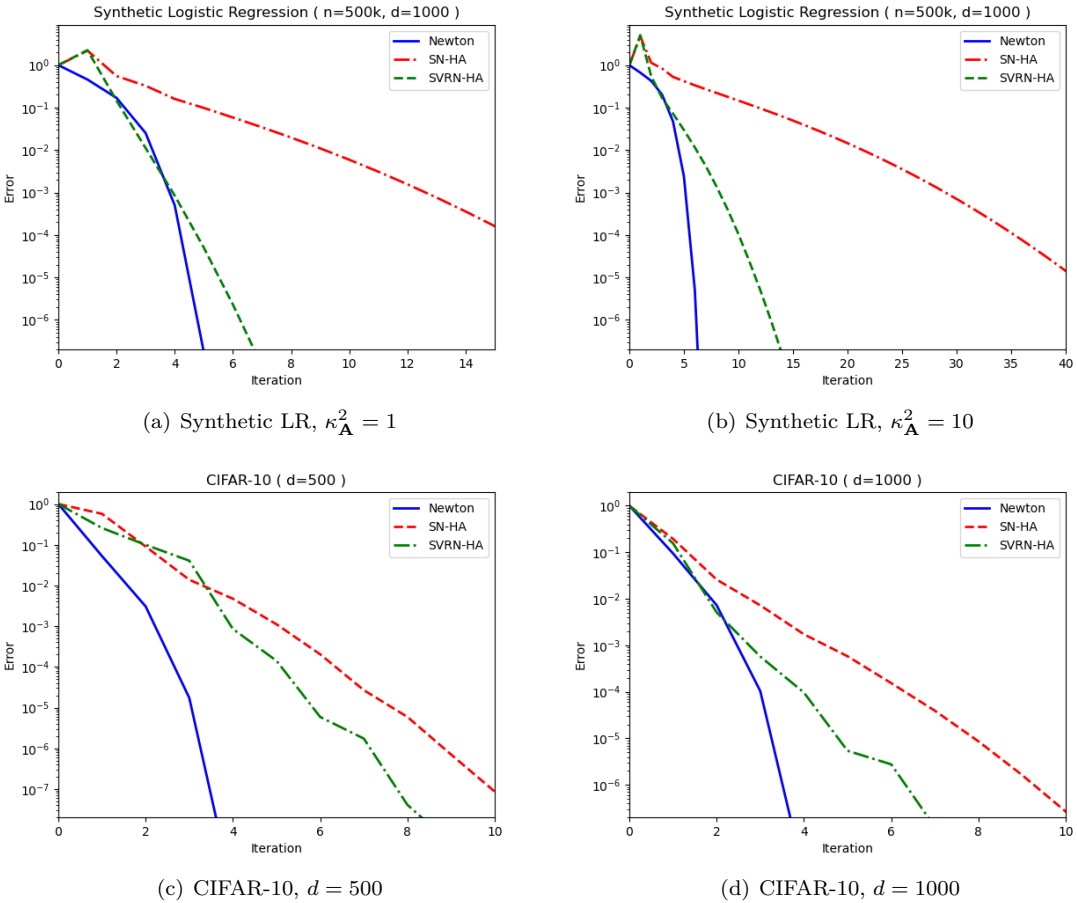

(a) Synthetic LR, $\kappa_{\mathbf{A}}^2 = 1$

(b) Synthetic LR, $\kappa_{\mathbf{A}}^2 = 10$

(c) CIFAR-10, $d = 500$

(d) CIFAR-10, $d = 1000$

Figure 5: Convergence comparison of SVRN-HA against SN-HA and Newton for a synthetic logistic regression task as we vary the condition number of the data matrix, and for the CIFAR-10 dataset.

## B  Further experimental details

In this section we provide additional details regarding our experimental setup in Section 5, as well as some further results on logistic regression with several datasets.

As a dataset, in Section 5, we used the Extended MNIST dataset of handwritten digits (EMNIST, Cohen et al., 2017) with $n = 500k$ datapoints. Here, we also include results on the CIFAR-10 image dataset with $n = 50k$ datapoints. Both datasets are preprocessed in the same way: Each image is transformed by a random features map that approximates a Gaussian kernel having width 0.002, and we partitioned the classes into two labels 1 and -1. We considered two feature dimensions: $d = 500$ and $d = 1000$, and we used the regularization parameter $\gamma = 10^{-8}$. To measure the error in the convergence plots, we use $\|\mathbf{x}_t - \mathbf{x}^*\|_{\mathbf{H}}^2 / \|\mathbf{x}_0 - \mathbf{x}^*\|_{\mathbf{H}}^2$, where $\mathbf{H} = \nabla^2 f(\mathbf{x}^*)$.

We next present further results, studying the convergence properties of SVRN on synthetic datasets with varying properties, for the logistic regression task as in (5). To construct our synthetic data matrices, we first generate an $n \times d$ Gaussian matrix $\mathbf{G}$, and let $\mathbf{G} = \mathbf{U}\mathbf{D}\mathbf{V}$ be the reduced SVD of that matrix (we used $n = 500k$ and $d = 1000$). Then, we replace diagonal matrix $\mathbf{D}$ with a matrix $\tilde{\mathbf{D}}$ that has singular values spread linearly from 1 to $\kappa_{\mathbf{A}}$. We then let $\mathbf{A} = \mathbf{U}\tilde{\mathbf{D}}\mathbf{V}$ be our data matrix. To generate the vector $\mathbf{y}$ for logistic regression, we first draw a random vector $\mathbf{x} \sim \mathcal{N}(\mathbf{0}, 1/d \cdot \mathbf{I}_d)$, and then we let $\mathbf{y} = \text{sign}(\mathbf{A}\mathbf{x})$.

For the least squares tasks in Section 5.2, we generated the same synthetic matrices $\mathbf{A}$, but with the vector $\mathbf{y}$ generated as follows: $\mathbf{y} = \mathbf{A}\mathbf{x} + \boldsymbol{\xi}$ where $\boldsymbol{\xi}$ is the Gaussian noise $\boldsymbol{\xi} \sim \mathcal{N}(\mathbf{0}, 0.1 \cdot \mathbf{I}_n)$. Here, we observed little difference in convergence behavior when varying $\kappa_{\mathbf{A}}$ (we show the results for $\kappa_{\mathbf{A}} = 10^3$).

**Logistic regression with varying condition number.** To supplement the EMNIST logistic regression experiments in Section 5, we present convergence of SVRN-HA on the CIFAR-10 dataset, as well as for the synthetic logistic regression task while varying the squared condition number $\kappa_{\mathbf{A}}^2$ of the data matrix. Note that, while $\kappa_{\mathbf{A}}^2$ is not the same as the condition number of the finite-sum minimization problem, it is correlated with it, by affecting the convexity and smoothness of $f$. From Figure 5, we observe that SVRN-HA outperforms SN-HA for both values of the data condition number. However, the convergence of both algorithms gets noticeably slower after increasing $\kappa_{\mathbf{A}}^2$, while it does not have as much of an effect on the Newton's method. Given that the increased condition number affects both methods similarly, we expect that the degradation in performance is primarily due to worse Hessian approximations, rather than increased variance in the gradient estimates. This may be because we are primarily affecting the global convexity of $f$, as opposed to the smoothness of individual components $\psi_i$. See our high-coherence least squares experiments for a discussion of how the smoothness of component functions affects the performances of SVRN and SN very differently.

## C  Related work on Subsampled Newton

In this section, we discuss several important prior works on Subsampled Newton methods to put our results in context. Specifically, we aim to illustrate how the Hessian approximation condition used in Theorems 1 and 3, i.e., $\frac{1}{\sqrt{\alpha}}\nabla^2 f(\mathbf{x}) \preceq \tilde{\mathbf{H}} \preceq \sqrt{\alpha}\nabla^2 f(\mathbf{x})$, relates to the Hessian estimates used in this line of works when showing fast local convergence rates. Also, in Appendix D.2, we show that to recover our condition with $\alpha \leq 2$ via uniform Hessian subsampling, one needs $O(\kappa \log(d))$ samples. Throughout this section, we use notation from the respective references.

First, we consider the Hessian averaging method studied by (Na et al., 2022). It is important to distinguish between the condition they impose on the stochastic Hessian oracle $\widehat{\mathbf{H}}$, and the Hessian approximation guarantee that they obtain for the actual estimate $\tilde{\mathbf{H}}_t$ that they use (and that we use in SVRN-HA). The stochastic oracle is only required to have a sub-exponential tail (see their Assumption 2.1, and also Example 2.3 illustrating this for Subsampled Newton). However, the actual estimate $\tilde{\mathbf{H}}_t$ is a result of averaging many samples from that oracle. Their local convergence analysis is only deployed once enough oracle samples are averaged so that $\tilde{\mathbf{H}}_t$ achieves the approximation guarantee given in their Lemma 3.5, i.e., $(1-\psi)\mathbf{H}_t \preceq \tilde{\mathbf{H}}_t \preceq (1+\psi)\mathbf{H}_t$. This approximation guarantee is strictly stronger than ours, but it becomes equivalent once $\alpha \leq 2$.

Next, we consider (Roosta-Khorasani & Mahoney, 2019) which analyzes a broad class of Subsampled Newton methods. In this paper, the most relevant results are Lemma 2 (Hessian approximation guarantee) and Theorem 5 (local convergence result). The guarantee reduces to our condition with $\alpha \leq 2$, with the only difference being that their Hessian approximation is restricted to the "cone of feasible directions", defined in (3). This restriction is only present in a constrained optimization setting (we focus on unconstrained optimization). The lemma also shows that the required Hessian sample size is again larger than the condition number of the problem (their condition number $\kappa_1$ matches our $\kappa$ for unconstrained optimization).

Next, we look at (Bollapragada et al., 2018) which presents a convergence analysis of Subsampled Newton under slightly different assumptions. Here, the key statements for local convergence are Lemma 2.4 in the journal version (Lemma 2.3 in arxiv), and equation (2.17). The lemma gives an approximation guarantee for the subsampled Hessian. This guarantee is in some sense weaker than our condition, because it only requires the Hessian approximation to be good in one direction, i.e., $w_k - w^*$ in their notation. However, examining the convergence bound in (2.17), for the local convergence analysis to hold, the Hessian sample size must still satisfy $|S_k| \geq \sigma^2/\bar{\mu}^2$ where $\sigma^2$ is effectively the upper bound on the component Hessians (potentially as large as our $\lambda$-smoothness) and $\bar{\mu}$ is the lower bound on the component Hessians. The latter is effectively a component-wise strong convexity constant, which can be much smaller (i.e., worse) than our global strong convexity $\mu$. In summary, their Hessian approximation condition for local convergence analysis, while slightly different, also requires the Hessian sample size to be larger than a condition number of the problem. Their condition number can be much larger than our condition number $\kappa$, or even infinite (for problems as simple as least squares), and is less standard in the literature.

Finally, we examine (Erdogdu & Montanari, 2015), where the authors consider Subsampled Newton with a possibly low-rank approximation of the Hessian. The most relevant result in that work is Lemma 3.1. Here, the standard version of Subsampled Newton is recovered when we let $Q^t = H_{S_t}^{-1}$. Then, the standard Hessian approximation condition appears implicitly through the fact that $\xi_1^t$ has to be less than 1 for the bound to be non-vacuous. To see the condition more clearly, we point to Equation (B.1) in the appendix (Equation A.1 in the arxiv version), which requires that $\|Q^t\| \cdot \|H_{S_t} - H\| < 1$. For $Q^t = H_{S_t}^{-1}$, this is essentially equivalent to our condition with $\alpha \leq 2$. Also, from the bound in Lemma 3.1, we see that once again the condition requires Hessian sample size to be larger than a condition number of the problem (which is for them $K\|Q^t\|$, and after some effort, this can be seen as comparable to our condition number). In the main algorithm of the paper, NewSamp, the authors aim to reduce the required sample size by using a different $Q^t$ computed from a low-rank approximation of $H_{S_t}$. This roughly corresponds to constructing a Hessian $\alpha$-approximation with $1 \ll \alpha \ll \kappa$.

## D  Omitted proofs

Here, we include the proofs of the auxiliary results stated in the paper. First, we discuss in detail the global convergence analysis of SVRN-HA (Theorem 4). Then, we illustrate how the Hessian approximation required in Theorem 3 can be obtained via subsampling.

### D.1  Global convergence of SVRN-HA

Here, we show how the proof of Theorem 4, i.e., global convergence of SVRN-HA (Algorithm 1), follows from global convergence analysis of Hessian averaging (Na et al., 2022). They show in Lemma 3.5 that if we were to run the global phase of SVRN-HA exclusively, then for any $\epsilon, \delta \in (0, 1)$ there is $T := T(\epsilon, \delta)$ such that with probability $1 - \delta$ for all $s \geq T$ we have $\tilde{\mathbf{x}}_s \in U_f(\epsilon)$, $\tilde{\mathbf{H}}_s \approx_\epsilon \nabla^2 f(\tilde{\mathbf{x}}_s)$, and $\eta_s = 1$. This means that, for any $\epsilon$, the probability that the above event does not happen with any $T < \infty$ is less than any $\delta > 0$, so it must be 0. This implies that SVRN-HA will eventually switch to the local phase (i.e., to SVRN). Note that it is possible that the switch will occur before the local neighborhood and Hessian approximation conditions are met. But if this causes SVRN to produce a poor descent direction, it will be caught by the line search (resulting in $\eta_s < 1$) and the method will simply revert back to the global phase. Eventually, the global phase will ensure that both conditions are met, and we can rely on Theorem 3 for the local convergence analysis.

### D.2  Hessian approximation via subsampling

Here, we illustrate how the Hessian $\alpha$-approximation condition (2), used in Theorems 1 and 3, can be obtained via uniformly subsampling $O(\kappa \log d)$ component Hessians. This result follows from Bernstein's concentration inequality for random matrices, given below (Tropp, 2012).

**Lemma 7 (Matrix Bernstein's inequality)** *Let* $\mathbf{Z}_1, ..., \mathbf{Z}_k$ *be independent random symmetric $d \times d$ matrices such that $\frac{1}{k} \sum_i \mathbb{E}[\mathbf{Z}_i] = \bar{\mathbf{Z}}$. Suppose that:*

$$\Big\| \frac{1}{k} \sum_i \mathbb{E}[(\mathbf{Z}_i - \mathbb{E}[\mathbf{Z}_i])^2] \Big\| \leq \bar{\sigma}^2 \quad and \quad \|\mathbf{Z}_i - \mathbb{E}[\mathbf{Z}_i]\| \leq R.$$

*Then, for any $\epsilon \geq 0$*

$$\Pr\Big( \Big\| \frac{1}{k} \sum_{i=1}^k \mathbf{Z}_i - \bar{\mathbf{Z}} \Big\| \geq \epsilon \Big) \leq 2d \cdot \exp\Big( -\frac{\epsilon^2 k/2}{\bar{\sigma}^2 + \epsilon R/3} \Big).$$

We are now ready to show the approximation guarantee for a subsampled Hessian estimate.

**Lemma 8** *Suppose Assumption 1, and let $\mathcal{D}$ be the sampling distribution for component functions $\psi$, as in Theorem 3. Let $\psi_1, ..., \psi_k \sim \mathcal{D}$ be i.i.d. samples from this distribution. There is an absolute constant $c$ such*

*that for any $\mathbf{x} \in \mathbb{R}^d$, with probability $1 - \delta$, the matrix*

$$\tilde{\mathbf{H}} = \left(1 + \frac{\gamma}{\mu}\right)^{-1/2}\left(\frac{1}{k}\sum_{i=1}^{k}\nabla^2\psi_i(\mathbf{x}) + \gamma\mathbf{I}\right), \qquad with \quad \gamma = \max\left\{12\lambda\log(2d/\delta)/k, \mu\right\},$$

*is an $\alpha$-approximation of $\nabla^2 f(\mathbf{x})$ as in* (2) *with*

$$\alpha = 1 + O\left(\kappa\log(d/\delta)/k + \sqrt{\kappa\log(d/\delta)/k}\right).$$

**Proof** Let $\mathbf{H}_\gamma = \nabla^2 f(\mathbf{x}) + \gamma\mathbf{I}$ for some $\gamma \geq 0$. We will use Lemma 7 with $\mathbf{Z}_i = \mathbf{H}_\gamma^{-1/2}\nabla^2\psi_i(\mathbf{x})\mathbf{H}_\gamma^{-1/2}$. First, note that $\mathbb{E}[\nabla^2\psi_i(\mathbf{x})] = \nabla^2 f(\mathbf{x})$ so that $\bar{\mathbf{Z}} = \mathbb{E}[\mathbf{Z}_i] = \mathbf{H}_\gamma^{-1/2}\nabla^2 f(\mathbf{x})\mathbf{H}_\gamma^{-1/2} \preceq \mathbf{I}$. Next, using that $\|\mathbf{H}_\gamma^{-1}\| \leq 1/(\mu + \gamma)$, we compute the boundedness parameter $R$:

$$\|\mathbf{Z}_i - \mathbb{E}[\mathbf{Z}_i]\| \leq \|\mathbf{H}_\gamma^{-1/2}\nabla^2\psi_i(\mathbf{x})\mathbf{H}_\gamma^{-1/2}\| + 1 \leq \frac{2\lambda}{\mu + \gamma} =: R.$$

Now, we similarly bound the variance parameter $\bar{\sigma}^2$:

$$\left\|\mathbb{E}[(\mathbf{Z}_i - \mathbb{E}[\mathbf{Z}_i])^2]\right\| \leq \left\|\mathbb{E}[\mathbf{Z}_i^2]\right\| \leq \left\|\mathbb{E}\left[\|\mathbf{Z}_i\|\mathbf{Z}_i\right]\right\| \leq \frac{\lambda}{\mu + \gamma}\|\mathbb{E}[\mathbf{Z}_i]\| \leq \frac{\lambda}{\mu + \gamma} =: \bar{\sigma}^2.$$

Thus, Lemma 7 implies that if $k \geq \frac{3\lambda}{\mu + \gamma}\log(2d/\delta)/\epsilon^2$ then with probability $1 - \delta$ the Hessian estimate $\tilde{\mathbf{H}} = \left(1 + \frac{\gamma}{\mu}\right)^{-1/2}\left(\frac{1}{k}\sum_{i=1}^{k}\nabla^2\psi_i(\mathbf{x}) + \gamma\mathbf{I}\right)$ satisfies:

$$\left\|\mathbf{H}_\gamma^{-1/2}\left(1 + \frac{\gamma}{\mu}\right)^{1/2}\tilde{\mathbf{H}}\mathbf{H}_\gamma^{-1/2} - \mathbf{I}\right\| = \left\|\frac{1}{k}\sum_{i=1}^{k}\mathbf{Z}_i - \bar{\mathbf{Z}}\right\| \leq \epsilon.$$

We can rewrite this as:

$$(1 - \epsilon)\mathbf{I} \preceq \mathbf{H}_\gamma^{-1/2}\left(1 + \frac{\gamma}{\mu}\right)^{1/2}\tilde{\mathbf{H}}\mathbf{H}_\gamma^{-1/2} \preceq (1 + \epsilon)\mathbf{I},$$

which is equivalent to:

$$(1 - \epsilon)\left(1 + \frac{\gamma}{\mu}\right)^{-1/2}\mathbf{H}_\gamma \preceq \tilde{\mathbf{H}} \preceq (1 + \epsilon)\left(1 + \frac{\gamma}{\mu}\right)^{-1/2}\mathbf{H}_\gamma.$$

Moreover, note that the regularized Hessian $\mathbf{H}_\gamma$ satisfies:

$$\nabla^2 f(\mathbf{x}) \preceq \mathbf{H}_\gamma = \nabla^2 f(\mathbf{x}) + \frac{\gamma}{\mu}\mu\mathbf{I} \preceq \left(1 + \frac{\gamma}{\mu}\right)\nabla^2 f(\mathbf{x}).$$

Putting this together, and assuming that $\epsilon \leq 1/2$, for $k \geq \frac{3\lambda}{\mu + \gamma}\log(2d/\delta)/\epsilon^2$ we get:

$$\frac{1}{\sqrt{\alpha}}\nabla^2 f(\mathbf{x}) \preceq \tilde{\mathbf{H}} \preceq \sqrt{\alpha}\nabla^2 f(\mathbf{x}), \qquad with \quad \alpha = (1 + 2\epsilon)^2\left(1 + \frac{\gamma}{\mu}\right).$$

Thus, if we set $\gamma = \max\{12\lambda\log(2d/\delta)/k, \mu\}$, then there are two cases:

1. If $k \leq 12\kappa\log(2d/\delta)$, then we have $k = 12\frac{\lambda}{\gamma}\log(2d/\delta) \geq \frac{3\lambda}{\mu + \gamma}\log(2d/\delta)/\epsilon^2$ for $\epsilon = 1/2$, which implies that $\tilde{\mathbf{H}}$ is an $\alpha$-approximation of $\nabla^2 f(\mathbf{x})$ with $\alpha = O(\kappa\log(d/\delta)/k)$.

2. If $k \geq 12\kappa\log(2d/\delta)$, then let $\epsilon = \sqrt{\gamma/2\mu}$, so that we have $k = 12\frac{\lambda}{\gamma}\log(2d/\delta) = 3\kappa\log(2d/\delta)/\epsilon^2$, which implies that $\tilde{\mathbf{H}}$ is an $\alpha$-approximation of $\nabla^2 f(\mathbf{x})$ with $\alpha = (1 + 2\epsilon)^3 = O(1 + \sqrt{\kappa\log(d/\delta)/k})$.

$\blacksquare$

