# OpenReview forum: "Stochastic Variance-Reduced Newton: Accelerating Finite-Sum Minimization with Large Batches"
_TMLR — Accepted by TMLR_

### Review · Reviewer_DCx8 · 2024-06-30

**Summary Of Contributions:**

The paper proposes a new algorithm, called Stochastic Variance-Reduced Newton (SVRN), for finite-sum minimization problems. It assumes we are minimizing the average of $n$ functions, and that this overall loss function is strongly convex and smooth (with condition number $\kappa$). Further, we assume that we can compute each of the $n$ functions, their gradients, and their Hessians.

The core claim of the paper is that, if we can access an $\alpha$-approximation to the Hessian (i.e. can access a matrix $\tilde H$ such that $\frac1{\sqrt\alpha} \ \nabla^2 f(\vec x) \lesssim \tilde H \lesssim \sqrt{\alpha} \ \nabla^2 f(\vec x)$), then SVRN converges using just $O(\alpha \log(1/\varepsilon))$ batches of gradient computations / rounds of adaptivity and $O(\frac{n}{\log(n)} \log(1/\varepsilon))$ total gradient computations. These bounds are local convergence bounds, assuming that we start iterating from a point $x$ not too far from the global comptimum $x^*$.

This rate matches the best known bound on rounds of adaptivity used in prior work, and outperforms the best known bound on total gradient computations by a factor of $\log(n)$.

The core premise in the algorithms is to integrate variance reduction techniques with a stochastic newton method.

A combination of both theoretical analysis and empirical evidence is provided.

**Audience:**

Yes

**Claims And Evidence:**

Yes

**Requested Changes:**

The bullet points with a $\star$ below are important to me. Most of these are fairly minor (add citations; clarify the meaning of a column in a table, etc). The bullet points without a $\star$ are recommendations and can be dropped.

- $\star$ [Page 2, Parallel Complexity at the bottom] add a citation to the standard PRAM model
- $\star$ [Page 2, Sequential Complexity at the bottom] add a citation to where this is used (see "nitpicks" above)
- $\star$ [Page 3, Table 1] clarify the speedup column. Is this a speed-up in sequential complexity? The speedup of SVRN wrt mini-batch SVRG and mini-batch Katyusha is entirely in the parallel complexity. The speedup of SVRN wrt stochastic newton is only in sequential complexity. This is kinda confusing to me overall. I don't really get what speedup you're talking about.
- $\star$ [Page 3, Table 1] would be nice to add some citations to which particular papers your getting these particular rates from (see "nitpicks" above wrt $\alpha$).
- $\star$ [Page 3, equation 2] Cite somewhere or otherwise argue that this notion of an $\alpha$-approximation is standard.
- [Page 3, Theorem 1] If $n > \frac1\varepsilon$, then the Sequential time becomes $<1$ passes over the data. Is there some lower order term that dominates the sequential complexity when $n > \frac1\varepsilon (i.e. moderate accuracy regime)? Or do you really perform less than one pass over the entire dataset?
- $\star$ [Page 4, below equation 3] Cite the $O(\alpha \log(1/\varepsilon))$ rate.
- $\star$ [Page 4, "In this setting" sentence] Doesn't sampling $k = O(\kappa \log(d))$ component Hessians get you back to the rates of SVRG? What's the upside here? The following sentence mentions a result of [Erdogdu Montanari] which uses far fewer samples. Can we get a value for $\alpha$ here which is much better than $\kappa$? There any theoretical rates you can place down to show this?
- $\star$ [Page 9, Figure 2] Add units for the wall clock time (seconds? minutes? hours?). Add confidence intervals over the multiple runs of the algorithm.
- $\star$ [Page 9, "All of the convergence plots" sentence] How many times did you run each experiment? That number matters.
- [Sections 5.1 and 5.2] Could you write down ballpark values of $\kappa$ and $\alpha$ for this problem? It's okay if it ain't possible, but it'd be nice to see what values we're really dealing with in relation to the preceding theory.
- [Page 11, Figure 4] Would be nice to also see the wall clock times for these experiments. Using a smaller Hessian or no preconditioning should make the iterations faster. At least that's my intuition. So, I'd really like to see how those all balance out. Can easily be in the appendix though.
- [Page 11, "Effect of Hessian Accuracy"] I'd really like to see the value of $\alpha$ in a plot or paragraph here. At the minimum, I'd like to see ballpark values for $\alpha$ attached to the trends in figure 4a. Preferably, I'd love to see a plot with something like $\alpha$ on the x-axis and "iterations needed for convergence" on the y-axis. Or perhaps a plot with iterations on the x-axis and $\alpha$ on the y-axis, with a different series for each value of $h$ used in Figure 4(a). In essence, I don't really feel like I've seen the real dependence on the accuracy of the Hessian. Instead, I've seen that different sample complexities can matter. These claims are not totally equivalent.

**Strengths And Weaknesses:**

The paper seems strong, and like a good fit for TMLR.

It proposes a simple algorithm with a clean motivation -- merging variance reduction with stochastic newton methods. My personal expertise is not specifically in netwon methods, so I cannot personally attest to the significance of this result relative to the prior work. That is, I am not specifically aware if there is or is not any other variance-reduced stochastic newton method. That said, taking the author's word for it, I feel that this core pitch is compelling.

The theorems are well written, and I have no reason to doubt their accuracy (though I did not verify the proofs in detail). The presentation is all around very solid. The experiments are interesting and carry a nice strong message. It's overall a good paper. I do have some nitpicks and points where I lack clarity, but overall I'm happy to recommend publishing this paper at TMLR.

Notably, both of my nitpicks can be resolved with cleaner citations to the prior work. In general, that's really my biggest issue with the paper. As someone who understands all of the words on the paper but isn't an expert in finite-sum minimization or stochastic newton methods, I don't know exactly where to look in the prior work to either A) fully appreciate the novelty of this result or B) verify that the assumptions made in their paper are standard or C) verify that their assertions about the prior work are correct. It's easily solved by adding more clear and direct citations.

---
### Nitpicks

I have two key technical points where I lack some clarity.

The first is about the metrics used in the paper. On the bottom of page 2, the authors discuss the two complexity metrics used in the paper: parallel complexity and sequential complexity. Parallel complexity is completely fine by me and makes a lot of sense (it's the number of "rounds of adaptivity" or "batches of gradient computations"). Sequential complexity feels a bit odd in the way it's phrased. They describe it as "passes over the data", but in a kinda weird way imo:
> One pass corresponds to $n$ queries of component gradients.

Which strikes me as kinda odd because it seems that n queries to the first gradient $\nabla \phi_1(x)$ would count as "1 pass over the data". That is, the sequential complexity seems to just be $\frac{\text{Total number of gradient computations}}{n}$, which doesn't sound a lot like "passes over the data" to me. This is why I phrased my "Summary of Contributions" in terms of the total number of gradient computations instead of the sequential complexity. A few sentences clearing up why it's okay to think of $n$ queries to $\nabla \phi_1(x)$ as a single pass over the data would be nice. Or perhaps just a citation to a paper or two where that particular definition of sequential complexity is used.


The second point where I lack clarity is in the $\alpha$-approximation to the Hessian. On page 3 in equation (2), the authors say that their analysis is built on having access to an approximation to the Hessian $\tilde H$ such that
$$
\frac1{\sqrt\alpha} \nabla^2 f(x) \lesssim \tilde H \lesssim \sqrt{\alpha} \nabla^2 f(x).
$$
This notion of an approximation slightly surprised me (namely, why is there a square root on the $alpha$ -- spectral approximations tend to just have a constant there without the square root). But the authors then show in Table 1 on page 3 that the prior work on Stochastic Newton methods does use this notion of $\alpha$-approximation (see the third row of the table, titled "Stochastic Newton"). I wanted to verify that the prior work uses this exact notion of $\alpha$-approximation, but I had a hard time finding this exact expression. I checked in [Roostra Mahoney], [Berhas et al], and [Derezinkski et al]. I would love to see either A) a clear citation to a paper which uses this exact notion of $\alpha$-approximation or B) a discussion of why this notion of $\alpha$-approximation is right.

---

> ### Author Response · Authors · 2024-08-12
> **Response to Reviewer DCx8**
>
> Thanks for the positive feedback and detailed comments. We will address all of the requested changes in the final version.
> - **Sequential complexity.**
> You're right, a "pass over the data" does not strictly mean that each component gradient is evaluated exactly once. On the other hand, since our main algorithms only rely on either computing full gradients or sampling component gradients uniformly at random, this means that one "pass" corresponds to evaluating each component gradient once *on average* (i.e., in expectation). We accept that this may be confusing and are happy to switch "sequential complexity" to refer to simply the number of component gradients.
> - **Hessian approximation condition.**
> We did provide a discussion (Appendix C) of how our notion of Hessian approximation compares with those of other works on approximate Newton methods, including [Roosta Mahoney], [Erdogdu Montanari], and others (unfortunately, there is no single standard notion). The main takeaway is that our condition captures several other conditions as special cases, and is relatively loose compared to prior work. We note that our choice to use the *square root* of $\alpha$ in the definition is merely a notational convention, designed so that the relative condition number of $\tilde H$ with respect to $H$ becomes $\alpha$. In fact our condition can be stated in an even more general yet equivalent way, by asking that $\beta_1\nabla^2 f(x)\preceq \tilde H \preceq \beta_2\nabla^2 f(x)$ for some fixed $0<\beta_1\leq \beta_2$. In this case, by simply rescaling the Hessian estimate, we observe that $\frac1{\sqrt{\beta_1\beta_2}}\tilde H$ is an $\alpha$-approximation in the sense of equation (2) for $\alpha = \beta_2/\beta_1$, which means that we can still apply our results and algorithms to any such Hessian estimates.
>
> - **Speed-up column.** The speed-up column in Table 1 is defined as the ratio between the parallel and sequential complexity. This reflects how much we gain from leveraging the finite-sum formulation via stochastic optimization, relative to just treating the problem as a generic minimization task. The way that the parallel and sequential complexities were defined, they will necessarily be equal as long as we are only using the full gradients $\nabla f(x)$. By sub-sampling the components, we can hope to make the sequential complexity (in terms of the "data passes") smaller than the parallel one.
> - **The case where $n \gg \log1/\epsilon$.**  In fact, the sequential complexity of our method does not go below 1 pass, since one has to compute the full gradient at least once, so strictly speaking this should say $O(1 + \frac{\log(1/\epsilon)}{\log n})$, or assume that $\epsilon$ is not exponentially small in $n$ (which is, arguably, a very extreme corner case).
> - **Sampling $O(\kappa\log d)$ component Hessians.**
> First, we should note that the $O(\kappa\log d)$ bound is very pessimistic, and in most cases where it makes sense to use second-order information, one can rely on much more efficient Hessian estimation (eg sketching as in Section 2.2). However, even in this worst-case sense, sampling $O(\kappa\log d)$ component Hessians does make sense compared to SVRG, since, once we are in the local neighborhood, a Hessian estimate constructed in this way becomes a valid $\alpha$-approximation for the rest of the optimization process, meaning that we do not need to refresh it at every outer iteration.
> - **Hessian estimate using fewer than $\kappa$ samples.** Hessian estimation using few component samples effectively amounts to constructing a low-rank approximation of the true Hessian. In particular, this is roughly the strategy used by [Erdogdu Montanari]. The actual quality of this aproximation (i.e., the parameter $\alpha$) ultimately depends on the spectral properties of the true Hessian (e.g., the rate of decay of its eigenvalues). However, a simple guarantee one can show (as an extension of Lemma 8 from Appendix D.2) is that from $k$ component samples we can construct an estimate $\tilde H$ with approximation factor $\alpha = 1+O(\kappa\log(d)/k + \sqrt{\kappa\log(d)/k})$.
> - **Experimental details.** The time units in Figure 2 are seconds, and we averaged all of the plots over 10 runs. The theoretical condition number $\kappa$ is a pessimistic notion of complexity, and we do not think this would be particularly indicative of empirical results. On the other hand, the Hessian approximation factor $\alpha$ should be a decent indicator of practical performance. Moreover, as discussed above, we expect it to be roughly inversely proportional to the Hessian sample size $h$ (and have observed this empirically). This is why we used $h$ as a proxy for Hessian accuracy in Figure 4a, taking into account that $\alpha$ is not a parameter that one would have direct access to in a practical setting. However, we are happy to also provide the values of $\alpha$ for Figure 4a in the final version, if the reviewer thinks this will be helpful.

---

### Review · Reviewer_X5u3 · 2024-09-20

**Summary Of Contributions:**

The paper "Stochastic Variance-Reduced Newton: Accelerating Finite-Sum Minimization with Large Batches" introduces Stochastic Variance-Reduced Newton (SVRN), a novel algorithm designed to accelerate finite-sum minimization tasks, particularly in the context of large datasets. The main contributions and new knowledge presented by the submission can be summarized as follows:
1. The authors propose SVRN, a finite-sum minimization algorithm that leverages variance reduction techniques to accelerate existing stochastic Newton methods. This is achieved by reducing the number of passes over the data from $O(\alpha log(1/\epsilon))$ to $O(\frac{log(1/\epsilon)}{log(n)})$, where $n$ is the number of sum components and $\alpha$ is the approximation factor in the Hessian estimate. This improvement is significant, especially for large datasets.
2. The paper provides a rigorous theoretical analysis of SVRN, demonstrating that it achieves a faster convergence rate compared to existing stochastic Newton methods. Specifically, the sequential complexity is improved by a factor of $O(\alpha log(n))$, while maintaining the same parallel complexity.
3. The authors propose a practical initialization method for SVRN by running a few iterations of a Subsampled Newton method with line search. This approach is shown to substantially accelerate Subsampled Newton.

**Audience:**

Yes

**Broader Impact Concerns:**

This paper primarily focuses on theoretical research and does not present any specific ethical concerns that need to be highlighted.

**Claims And Evidence:**

Yes

**Requested Changes:**

1. About the communication cost: It would be better to provide a detailed discussion on the trade-offs between theoretical guarantees and practical efficiency, particularly regarding the frequency of gradient resampling, and explain the impact of resampling per stage versus per step.
2. About Hessian sample size: I think conducting a sensitivity analysis of the Hessian sample size, especially in the initial stages of the algorithm is important. And it would be better to provide guidelines on how to choose an appropriate Hessian sample size for different types of problems and datasets.

**Strengths And Weaknesses:**

Strengths:
1. The introduction of SVRN is a significant contribution, as it combines the benefits of variance reduction with second-order information, leading to faster convergence rates for finite-sum minimization tasks.
2. The paper provides a thorough theoretical analysis of SVRN, proving that it accelerates the convergence rate of existing stochastic Newton methods.
3. The proposed initialization method using a few iterations of a Subsampled Newton method with line search is practical and effective. This initialization significantly accelerates the convergence of Subsampled Newton.

Weaknesses:
1. While the theoretical analysis requires fresh samples of components for each small step, the practical implementation suggests that resampling per stage can be more efficient. The authors should provide a more detailed discussion on the trade-offs between theoretical guarantees and practical efficiency in terms of communication cost.
2. The performance of SVRN is sensitive to the Hessian sample size, especially in the initial stages. The authors should provide more insights into how to choose an appropriate Hessian sample size for different types of problems and datasets.

---

> ### Author Response · Authors · 2024-09-24
> **Response to Reviewer X5u3**
>
> We thank the reviewer for the positive feedback and questions. We will add the requested discussions in the final version.
>
> **Communication cost of gradient resampling**
>
> We agree that the frequency of gradient resampling has a significant impact for practical implementations of SVRN, particularly in terms of the communication cost (see our discussion in Section 5.2 and Figure 3a). The specific trade-offs in communication cost between the theoretical and practical variants of the algorithm are largely dependent on the implementation and architecture details, e.g., whether all of the data is stored on one machine or not, whether we have random access or streaming access, etc. As an example, let us consider the setting where each full/mini-batch gradient computation requires reading the corresponding data chunk from the server onto the computing core. Then:
>
> - The theoretical version of SVRN (sampling per step) will require (the equivalent of) reading the entire dataset $2$ times (i.e., $2n$ data points) in the course of one stage (outer iteration): $n$ data points for computing the full gradient, and then $t_{\max}\cdot m = (n/m)\cdot m = n$ data points for all of the mini-batch steps together.
>
> - The practical version of SVRN (sampling per stage) will require reading the entire dataset $1+o(1)$ times (namely, $(1+\frac1{\log(n/d)})\cdot n$ data points, i.e., asymptotically one data pass) in the course of one stage (outer iteration): $n$ data points for computing the full gradient, and then $1\cdot m = \lceil n/\log(n/d)\rceil$ data points for a single mini-batch reused in all $t_{\max}$ steps.
>
> This difference could be made even more significant by the fact that computing the full gradient can be done via streaming access to the data, whereas computing the random mini-batches requires (at least in theory) random access. All of these reasons are why we recommended sampling per stage as the practical implementation of SVRN, given that it does not appear to suffer in terms of the convergence rate as shown on Figure 3a (this is also the primary implementation we used in the experiments). Providing theoretical convergence analysis for this variant of SVRN is a very interesting direction for future research, as it will require deviating even further from the typical stochastic gradient-type analysis.
>
> **Sensitivity to Hessian sample size**
>
> The choice of Hessian sample size is indeed very important for SVRN. We illustrate this in Figure 4a, comparing SVRN and SN with different Hessian sample sizes (see discussion on pages 11-12). This can be addressed in one of two ways:
>
> - First, as shown in Theorem 3, SVRN can adapt to different values of Hessian accuracy $\alpha$ (which is inversely proportional to the Hessian sample size $k$ via $\alpha = \tilde O(1 + \kappa / k)$) by adjusting the step size $\eta$ (in Theorem 3, we used $\eta = \min(\sqrt{2/\alpha},1)$). However, since $\alpha$ is not readily available at runtime, this would require having to tune the step size, just like one has to do for SVRG.
>
> - Second approach, which avoids any hyper-parameter tuning, is our algorithm SVRN-HA (Algorithm 1). This approach uses Hessian Averaging to gradually improve the Hessian accuracy $\alpha$, until we can use SVRN with unit step size $\eta=1$. While Algorithm 1 technically still has a Hessian sample size parameter $k$, the method is much more robust to this parameter than a typical Sub-sampled Newton-type method, since no matter what $k$ we choose, the averaged Hessian will eventually become accurate enough for SVRN to work with unit step size (Theorem 4). This is why we argue that our simple recommendation of $k=4d$ in SVRN-HA should be a safe choice for this method across different problems and datasets (we supported this with experiments on two different problems and three datasets).

---

### Review · Reviewer_8WgW · 2024-10-02

**Summary Of Contributions:**

This work proposes SVRN (Stochastic Variance-Reduced Newton) for finite-sum minimization of smooth strongly convex functions.

Assumptions:
* Specifically, it is assumed that the global objective is strongly convex and the component objectives are each L-smooth and convex.
* It is further assumed that the objective is twice continuously differentiable and is either self-concordant, or has a Lipschitz Hessian.

Method:
* The proposed approach is to first commence with a variant of Subsampled Newton (Na et al., 2022), where the full batch gradient must be computed in each iteration, and the Hessian estimates are produced by random sampling of individual components, and Hessian Averaging across time to improve the estimates (i.e.; directly leveraging Subsampled Newton with Hessian Averaging; SN-HA).
* Once SN-HA has converged to a sufficiently small neighborhood of the minimizer (as determined by the SN-HA Armijo line-search), the proposed SVRN method kicks in.
  * The proposed SVRN method, which only operates in the local neighboorhood of the minimizer, is essentially SVRG, scaled by the inverse Hessian estimate, which is computed using random sampling and Hessian Averaging.

Results:
* It is proven that SN-HA proposed in previous work will indeed converge to a sufficient local neighborhood of the minimizer, such that SVRN will be able to kick in.
* It is also proven that, given sufficiently accurate Hessian estimates, SVRN will achieve accelerated local convergence to the global minimizer. Specifically, the parallel time (number of parallel batch queries) for SVRN is shown to be similar to Stochastic Newton, while Sequential time (number of passes over the dataset) is reduced by a factor of ~log(n).
* Numerical results are provided on least squares and logistic regression tasks, comparing both the iteration-wise and wall-clock convergence of a) full-batch Newton, b) Subsampled Newton with Hessian Averaging, c) SVRG, and d) SVRN (where SN-HA is used to enter the local neighborhood). Ablations are conducting in the least squares setting examining the effect of leverage scores on the proposed approach.

**Audience:**

Yes

**Broader Impact Concerns:**

N.A.

**Claims And Evidence:**

Yes

**Requested Changes:**

* Please add comparisons to accelerated variance-reduced methods in the numerical results, both momentum-based, such as Katyusha, and with inexact preconditioning.
* It is not clear to me why SN-HA and SVRN-HA have non-identical convergence early on in training (Figure 2).
* Please consider annotating the points in the plots at which the SVRN update kicks in, and the SN-HA updates halt.
* Please elaborate on how the problem conditioning still impacts the derived theory; ideally second order methods should be robust to problem conditioning, which is reflected in your rates, but somehow still appears to pop out from the bounds on the required hyper-parameters.
* While a main motivation of this work is explaining how SVRN improves local convergence with large batch sizes, it is not clear to me in the exposition or the numerical results where the improved dependence on batch size comes in relative to momentum accelerated variance-reduction methods.

**Strengths And Weaknesses:**

Strengths:
* The exposition is relatively clear; the paper is sufficiently well written.
* The proposed method is conceptually simple to understand and a simple extension combining SVRG and Subsampled Newton.
* The assumptions are relatively standard for second order methods.
* The theoretical results demonstrate improved local convergence compared to vanilla subsampled newton.
* Numerical results provide convincing evidence that the proposed SVRN improves local convergence of Subsampled Newton, while also notably reducing wall clock time relative to full-batch Newton.
* Some limitations of the method are discussed; ablations on leverage scores in linear least squares highlight the sensitivity of the proposed approach to subsampling for building sufficiently accurate Hessian estimates, where the number of sampled components needed to produce sufficiently accurate estimates will depend on the component-wise smoothness of the objective.

Weaknesses:
* The motivation for the proposed approach and comparison to related work could be better articulated. Why would inexactly preconditioned SVRG (e.g., Liu et al., 2019) not yield a similar effect as the proposed SVRN (improving local convergence with large mini-batches). Some positives; the comparison to full-batch second order methods is clear (SVRN using stochastic estimates). Moreover, the comparison to subsampled second order methods is clear (they don't have variance reduction). The comparison to certain accelerated variance-reduction methods is also somewhat clear; some of these approaches rely on momentum (such as Katyusha).
* While the proposed local convergence theory has removed the dependence on the condition number relative to accelerated variance-reduced methods, the minimum mini-batch size required for local accelerated convergence still depends on the condition number of the objective.
* Novelty is somewhat limited, initial phase is just subsampled newton, latter phase is SVRG using the hessian estimates from subsampled newton; however, as novelty is not a TMLR criterion, this will not affect by judgement of this work.

---

> ### Author Response · Authors · 2024-10-07
> **Response to Reviewer 8WgW**
>
> Thanks for the feedback. We will include the requested changes in the final version.
>
> **Main novelty.**
>  We agree that the algorithm SVRN (and SVRN-HA) is a natural combination of SVRG with Subsampled Newton-type preconditioning. The main novelty of this paper is to provide a new convergence analysis for this type of algorithm, which allows for large gradient mini-batches and high-probability guarantees. These types of guarantees are novel in SVRG literature. Prior works on preconditioned SVRG (eg, Liu et al, see below) rely on convergence analysis that closely follows the classical SVRG approaches (single-sample gradients, global analysis, in expectation), and they cannot recover our Newton-type guarantees.  In addition to demonstrating the effectiveness of large-batch gradients for SVRN, our analysis also enables a direct comparison with second-order methods such as Subsampled Newton, showing a clear benefit in using variance reduction even for gradient estimates that are already very accurate, which goes against popular wisdom in the area.
>
>
> **Comparison with Liu et al (2019).**
> Algorithmically, the Inexact Preconditioned SVRG studied by Liu et al (2019) is indeed closely related to SVRN. The main differences are: they do not allow for large mini-batches, and they make much stronger assumptions on the loss which allows them to use a single global preconditioner matrix $M$. The main novelty of our paper is the convergence analysis, which allows us to handle a more general class of problems and much larger gradient mini-batches. Concretely, we show our results under standard Lipschitz Hessian assumption, which forces us to perform a local convergence analysis which is very different from standard SVRG analysis. On the other hand, Liu et al assume that the global smoothness and strong convexity constants of the loss are measured in the norm induced by the preconditioner $M$, so that they can replicate the standard SVRG analysis under a different norm (they even state that their convergence Theorems 1 and 2 follow similarly to prior works). If one were to adapt their algorithm to handle large gradient mini-batches and a subsampled Hessian preconditioner, then the main difference from SVRN would be that they use an inexact inner solver for applying the preconditioner. One could also easily incorporate an inexact inner solver into SVRN, and adapt our analysis to allow for that, similarly to how it is done for Sub-sampled Newton methods (e.g., Roosta-Khorasani and Mahoney, 2019).
>
> **Benefits of large mini-batch size vs momentum acceleration.**
> The benefits of using large mini-batch size come from the fact that on most computing architectures it takes far less time to compute a single gradient estimate on a batch of $m$ component functions at one location, than it takes to compute $m$ component gradients at $m$ different locations, one at a time. We highlight this by the notion of parallel time, which measures the number of batch gradient queries. However, the benefits of large mini-batch sizes are not limited to parallel computing. Even standard single core architectures benefit substantially from the vectorization of gradient computations, which is only effective when using large batches. We observed this in our empirical runtime comparisons between SVRG and SVRN. We note that momentum acceleration cannot address this issue, because (as seen in Table 1) a momentum accelerated first-order method is still a first-order method, so it has to do at least $\Omega(\sqrt\kappa\log1/\epsilon)$ batch gradient queries, according to known lower bounds. We are happy to add to the final version an empirical comparison illustrating this (e.g., with Katyusha).
>
> **Dependence on the condition number.**
> The dependence on the condition number $\kappa$ in our results comes from sub-sampled gradient estimation. This is consistent with Subsampled Newton works such as Roosta-Khorasani and Mahoney (2019), where, to recover their fast condition number-free convergence guarantees they require the gradient sample size to be sufficiently larger than $\kappa$.  We showed how this can be avoided for certain losses (least squares, see Theorem 2 and Lemma 6) by relying on importance sampling.
>
> **Transition between SN and SVRN in SVRN-HA.**
> In our experiments, the transition point between SN and SVRN occurred very quickly, generally within 1-2 iterations. This is also consistent with the trajectory of the Newton's method, which appears to reach quadratic convergence within no more than 2 iterations, suggesting that the logistic regression objective possesses good Hessian smoothness properties. Any differences between SN-HA and SVRN-HA in the plots before the transition point are due to the random noise coming from Hessian sub-sampling, since each curve is generated from different runs of the algorithms.

---

### Author Response · Authors · 2024-10-13
**Response and Discussion**

Dear Reviewers,

Thank you again for your positive feedback and questions. We have responded to each review individually.
As the author-reviewer discussion period approaches the end, we wanted to check whether you have any further questions or concerns that we can address?

Best regards,
Authors.

---

### Decision · Action_Editor_mr5b · 2024-12-19

**Recommendation:** Accept with minor revision

**Comment:**

Authors already responded to the reviews thoroughly. There are a few minor issues that remain to be addressed.

- please include a discussion on the effect of condition number on the minimum mini-batch size (which was also raised by other reviewers).
- please clarify the numerical results in Figure 2 that the SVRN update kicks in after only a couple of iterations, which also explains why the SN-HA and SVRN updates are not identical in the early phases of the plot.
- please emphasize the motivation of the work in the introduction.
- please discuss differences between theoretical guarantees and practical efficiency (e.g. in terms of communication cost).

Once these are addressed, the paper will be ready for publication at TMLR.

**Audience:**

Yes, the paper is a good fit for TMLR audience.

**Claims And Evidence:**

Three reviewers reviewed the paper and they all found the claims accurate.